

# Observed Impacts of Aerosol Regimes on Energy and Carbon Fluxes in the Amazon Forest

Mariano A.B. da Rocha[1], Cléo Q. Dias-Júnior[1, 2, 3], Julia C.P. Cohen[1], Flávio A.F. D'Oliveira[2], Anne C.S. Mendonça[3], Christopher Pöhlker[4], Subha Raj[4], Alessandro C. de Araujo[1, 5], Marco A. Franco[6], Paulo Artaxo[7], Carlos A. Quesada[8], and Rafael S. Palácios[9]

[1]Graduate Program in Environmental Sciences, Federal University of Pará, Belém, Pará, Brazil
[2]Department of Physics, Federal Institute of Pará, Belém, Pará, Brazil
[3]Graduate Program in Climate and Environment, National Institute of Amazonian Research, Manaus, Amazonas, Brazil
[4]Multiphase Chemistry Department, Max Planck Institute for Chemistry, Mainz, Germany
[5]Empresa Brasileira de Pesquisa Agropecuária, Belém, Brazil
[6]Department of Atmospheric Sciences, Institute of Astronomy, Geophysics and Atmospheric Sciences, University of São Paulo, São Paulo, Brazil
[7]Institute of Physics, University of São Paulo, São Paulo, Brazil
[8]National Institute of Amazonian Research, Manaus, Amazonas, Brazil
[9]Institute of Geosciences and Faculty of Meteorology, Federal University of Pará, Belém, Pará, Brazil

**Correspondence:** Cléo Q. Dias-Júnior (cleo.quaresma@ifpa.edu.br)

**Abstract.** Atmospheric aerosols play a crucial role in modulating the energy available to the Earth's surface, influencing the hydrological cycle, ecosystems, and climate. In the Amazon, previous studies have mainly examined how aerosols scatter and absorb radiation, enhancing diffuse radiation and influencing gross primary productivity. However, little is known about their interactions with energy partitioning (i.e., sensible and latent heat fluxes). Here, we investigate how regimes of high (AOD > 0.40) and low (AOD < 0.13) aerosol optical depth (AOD) affect surface energy and carbon dioxide ($CO_2$) fluxes in an undisturbed Amazon rainforest. For this, we used long-term meteorological measurements from the Amazon Tall Tower Observatory (ATTO) collected between 2016 and 2022. We find that enhanced aerosol presence reduces both sensible heat flux and energy available for evapotranspiration by approximately 10%, while decreasing $CO_2$ fluxes by about 58%, which suggests enhanced carbon assimilation by the forest. The impact of aerosols on turbulent surface fluxes is reflected in a cooling of approximately 0.5 °C at the canopy top, caused by a 5.6% reduction in incoming shortwave radiation. These results demonstrate that aerosols modify turbulent energy exchange, with consequences for the forest microclimate and the coupled carbon and water cycles. It highlights the critical role of aerosols in the functioning of the ecosystem.

## 1 Introduction

Atmospheric aerosols, which are defined as solid or liquid particles suspended in the air (Seinfeld and Pandis, 2006), play a multifaceted role in the Earth system. They influence the atmospheric cycle (Lohmann and Feichter, 2005; Rap et al., 2013; Gavrouzou et al., 2023), the hydrological cycle (Miller et al., 2004; Lau et al., 2005; Suzuki et al., 2017), and ecosystem processes (Kanakidou et al., 2018; Artaxo et al., 2022; Karthick Raja Namasivayam et al., 2024).





In the atmosphere, aerosols interact directly with solar radiation through scattering and absorption processes. These interactions influence the Earth's energy balance and, consequently, the climate (Liu et al., 2020). Aerosols also act indirectly by
interacting with clouds, acting as cloud condensation nuclei (CCN). This interaction alters the albedo, formation, microphysics, and lifetime of clouds, thereby impacting global climate patterns (Andreae et al., 2004; Eltbaakh et al., 2012; Wang and Yi, 2024).

In the hydrological cycle, aerosols reduce the intensity of precipitation through complex, partially nonlinear processes that involve suppression of convection through mechanisms of aerosol-radiation interaction that stabilize the atmosphere, particu-
larly at levels of aerosol optical depth (AOD) greater than 0.40 (Herbert and Stier, 2023). This results in a greater number of cloud droplets with a radius of less than 14 $\mu$m forming, which are insufficient for precipitation (Ramanathan et al., 2001; Gonçalves et al., 2015). In addition, they influence downdrafts, altering the concentration of gases near the surface (D'Oliveira et al., 2022). Aerosols also reduce global evapotranspiration, which has a more significant impact on tropical forests (Liu et al., 2014).

In forest ecosystems, high concentrations of aerosols can increase the intensity of diffuse radiation, which positively impacts photosynthetic rates (Li et al., 2025). This phenomenon, known as diffuse fertilization, mainly benefits shaded areas, allowing them to carry out photosynthesis more efficiently (Kanniah et al., 2012).

The Amazon region, home to the world's largest rainforest, has been the site of significant research on the intricate relationship between aerosols, the biosphere, the atmosphere, and human activities. Since the 1980s, several scientific projects have
been conducted in the region to better understand these interactions Harriss et al. (1988); Avissar et al. (2002). Other studies have deepened our knowledge of the formation, transformation and impact of aerosols, particularly on clouds and precipitation (Yokelson et al., 2007; Martin et al., 2010; Brito et al., 2014; Machado et al., 2014; Martin et al., 2017). The Amazon Tall Tower Observatory (ATTO) project has recently played an instrumental role in monitoring long-term changes and in understanding the role of aerosols in global climate and the Amazon ecosystem (Andreae et al., 2015).

Aerosols in the Amazon are mainly composed of organic carbon, accounting for more than 80 % of their mass (Artaxo et al., 2013). This proportion varies seasonally and can exceed 90 % during the burning seasons (Artaxo et al., 2013). During the wet season, aerosol concentrations are low and similar to those of concentrations above the ocean. However, in the dry season, fires drastically increase the aerosol load, which affects cloud formation and precipitation (Andreae et al., 2004). These particles also alter the radiative balance, significantly affecting carbon absorption by the forest Rodrigues et al. (2024). Changes in land
use and an increase in fires not only lead to higher levels of pollution, but also reduce rainfall efficiency and modify the regional climate. This creates a positive feedback loop that can result in two different climatic states: one humid and sparsely polluted and the other dry and highly polluted (Andreae et al., 2004; Pöhlker et al., 2019).

Despite advances in understanding aerosol-biosphere-atmosphere interactions in the Amazon, the impact of these particles on energy and radiation partitioning is still unclear. Using numerical simulations for the Amazon basin, Braghiere et al. (2020)
showed that there are considerable uncertainties about the influence of aerosols on the surface energy balance. Their simulations also revealed that, in a scenario without aerosols (AOD = 0), the sensible and latent heat fluxes were higher than those measured experimentally, resulting in higher surface temperatures. Furthermore, recent studies, such as those by Blichner et al. (2024),





reveal that numerical models still fail to accurately portray the interaction between aerosols and thermal effects in the Amazon. This is mainly due to the models' inability to adequately capture the relationship between temperature and organic aerosol concentrations.

The aim of this study was to evaluate the influence of aerosols on energy fluxes (net radiation – Rn, sensible heat – H, and latent heat – LE) and mass flux (carbon dioxide flux – $FCO_2$) at the forest-atmosphere interface in an undisturbed region of the Amazon. Using in situ measurements, the study analyzed the period between 2016 and 2022, contributing to our understanding of processes involving the interaction between atmospheric aerosols and the energy balance in an area of pristine Amazon forest. To date, we are unaware of any studies that have used experimental data measured close to the surface to examine the relationship between aerosols and energy partitioning in the Amazon.

## 2 Material and Methods

### 2.1 Experimental site

The data used in this study were collected as part of the ATTO project, a bilateral initiative between Brazil and Germany. Since 2012, ATTO has carried out continuous measurements, as described by Andreae et al. (2015), located in an area of pristine tropical forests in the central Amazon (Figure 1), which contains several measurement towers, including the Instant Tower of 81 meters (-2.1441° S, -58.9999° W) and the Tall Tower of 325 meters (-2.1459° S, -59.0056° W).

The ATTO towers are located 150 km from the city of Manaus in the state of Amazonas, Brazil, at an altitude of 120 meters above sea level on a plateau covered by terra firme forests with an average crown height of 40 meters (Gomes Alves et al., 2023). In this landscape, wind speeds are relatively low, around 1 m/s immediately above the forest canopy, and above the canopy, the wind speed increases logarithmically with height (Santana et al., 2016). The main wind direction at the site is from the NE – E. It passes through areas of minimal anthropogenic influence in the northeast, a clean fetch region covered by tropical forests (Pöhlker et al., 2019).

The climate is tropical humid, divided into two seasons (wet and dry) with the predominant origins of the air masses due to the large seasonal shifts of the Intertropical Convergence Zone over the Amazon Basin (Andreae et al., 2015). The wet season is characterized by more than 200 mm of rainfall per month and an average temperature of around 25° C at the forest-atmosphere interface. In contrast, the dry season sees less than 100 mm of rainfall per month and an average temperature of around 27.7° C (Schmitt et al., 2023).

### 2.2 Experimental data

The dataset used in this study was measured at the ATTO site from 2016 to 2022 (see Table 1). Wind speed (ws), sensible heat flux (H), latent heat flux (LE), and $FCO_2$ data were calculated as 30-minute averages using EddyPro® software (LiCor), as derived from fast-response sonic anemometers, according to Fratini and Mauder (2014). The other variables (radiation, thermodynamics and aerosols) were obtained as 30-minute averages.





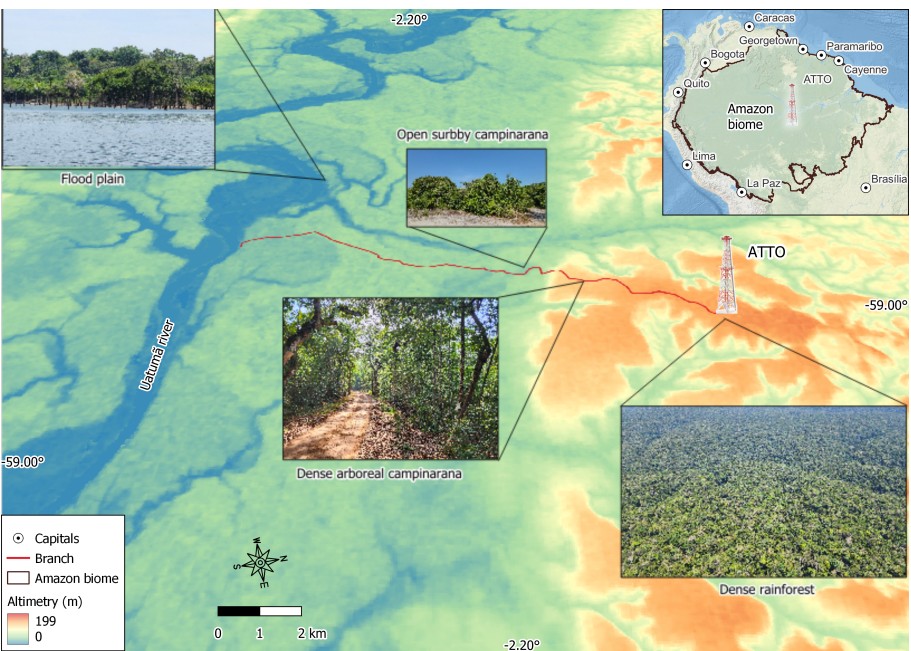

**Figure 1.** Amazon Tall Tower Observatory (ATTO) in central Amazonia, which has different landscapes along the topographic gradient, including floodplains, shrubby campinarana, dense arboreal campinarana, and dense ombrophilous forests. It is close to the Uatumã River, which runs in an NW-SE direction and is a tributary of the left bank of the Amazon River. Altimetry data by NASA JPL (2020) and vetorial data by RAISG (2023).

Based on Andreae et al. (2015) and Pöhlker et al. (2016), these data were organized by seasonality into four periods: (i) the wet season (February to May), which has a cleaner atmosphere, (ii) the wet-dry transition (June to July), (iii) the dry season (August to November), which has higher levels of pollution, and (iv) the dry-wet transition (December to January).

Initially, the database containing the variables in Table 1 had 10,890 rows, each of which has values averaged over 30 minutes. However, some filters were applied: i) the turbulent fluxes underwent quality control based on Foken et al. (2004). Only data with flags "0" (best quality) and "1" (acceptable for general analysis) were used; data with flag "2" (poor quality) were discarded; ii) the presence of clouds interferes with radiation balance and energy partitioning. To eliminate cloud interference and investigate the role of aerosols in surface energy fluxes, the central objective of this study, data from the AERONET (Aerosol Robotic Network) at the ATTO site, specifically AOD (version 3, level 2). These data are free of cloud contamination due to pre and post-field calibration (Giles et al., 2019); iii) this study only considered the daytime period (from 6:00 to 17:00 LT) because the highest Rn values occur during this time. After filtering, the database was reduced to 523 rows: 370 in the dry season and 153 in the wet season. Of the filtered data, 2020 had the highest availability at 42.4 %, followed by 2022 at 29.2%, then 2016 at 20 %, 2021 at 4.9 %, 2019 at 3 %, and finally 2017 at only 0.5 %, however, no data was available for 2018 (0%). This distribution reflects the difficulty of obtaining robust data sets with which to assess the effects of aerosols.

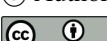



**Table 1.** Variables and the methods used to obtain them.(*) Calculations according to Bolton (1980).

| Type of Variable | Variable | Unit | Hight(m) | Method of production | Data sampling rate |
|---|---|---|---|---|---|
| Radiation | Short Wave Radiation (SW) | $\mathrm{W\,m^{-2}}$ | 75 | Kipp&Zonen CMP21 | 1 min |
| | Long Wave Radiation (LW) | $\mathrm{W\,m^{-2}}$ | 75 | Kipp&Zonen CGR4 | 1 min |
| | Net Radiation (Rn) | $\mathrm{W\,m^{-2}}$ | 75 | Kipp & Zonen NR-LITE2 | 1 min |
| Thermodynamics | Air temperature (T) | °C | 80 | GALLTEC-MELA IAK I-Series | 1 min |
| | Relative humidity (RH) | % | 80 | GALLTEC MELA IAK I-Series | 1 min |
| | Air Pressure (Patm) | hPa | 80 | YOUNG 61302V | 1 min |
| | Wind speed (ws) | $\mathrm{m\,s^{-1}}$ | 80 | CSAT3B & THIES 4.3830 | 1 min |
| | Vapor pressure deficit (VPD) | hPa | - | Calculation* | 1 min |
| | Mixing ratio (r) | g of vapor/kg of dry air | - | Calculation* | 1 min |
| | Soil temperature (Ts) | °C | 0.1 | Campbell Thermistor 108 | 10 min |
| | Soil moisture (h) | $\mathrm{m^3\,m^{-3}}$ | 0.1 | Campbell CS615 | 10 min |
| Flux | Sensible Heat (H) | $\mathrm{W\,m^{-2}}$ | 80 | CSAT3B/LI-7200RS | 10 Hz |
| | Latent Heat (LE) | $\mathrm{W\,m^{-2}}$ | 80 | CSAT3B/LI-7200RS | 10 Hz |
| | Carbon dioxide (FCO2) | $\mathrm{\mu mol\,m^{-2}\,s^{-1}}$ | 80 | CSAT3B/LI-7200RS | 10 Hz |
| | Soil heat (G) | $\mathrm{W\,m^{-2}}$ | 0.05 | Hukseflux HFP01 | 10 min |
| Aerosols | Aerosol Optical Depth 500 nm (AOD) | - | 80 | CIMEL Sun Photometer CE318-T | Variable rate |

Using the values for humidity, temperature and atmospheric pressure (variables shown in Table 1), it was possible to calculate the vapor pressure deficit (VPD) and the mixing ratio ($r$) using Equations 1 and 4. These calculated and directly measured variables in Table 1 allowed the Random Forest model (explained in Section 2.3) to be used to calculate the impact of aerosols on the energy and matter fluxes.

VPD was calculated according to the equations used by Bolton (1980).

$$VPD = e_s - e_a \tag{1}$$

The water vapor saturation pressure ($e_s$) as a function of temperature was calculated according to the equation Tetens (1930).

$$e_s(T) = 6.112 exp(\frac{17.67 \cdot T}{T + 243.5}) \tag{2}$$

The actual vapor pressure ($e_a$) was obtained by relating it to the relative humidity (RH).

$$e_a = RH \cdot e_s \tag{3}$$

$r$ was calculated using a process that satisfies the equivalent potential temperature equation, as described by Holton (2013). In this equation, $P_{atm}$ represents the atmospheric pressure.

$$r = \frac{622 \cdot e_a}{P_{atm}} - e_a \tag{4}$$

### 2.3 Analysis methods

Initially, daily averages of AOD values were obtained to investigate seasonal variability. The days with extreme AOD values were then selected for which hourly averages were obtained between 06:00 and 17:00 LT. To explore how extreme AOD



conditions influence surface turbulent fluxes, a 4th-order polynomial adjustment was used, similar to that carried out by Meyers

and Dale (1983).

This analysis enabled the behavior of the variables to be investigated under two extreme atmospheric conditions: the 'polluted' and 'clean' regimes in the ATTO region, which are characterized by high and low aerosol loads (AOD > 0.40 and AOD < 0.13, respectively).

Statistical analyzes were also used: i) Spearman's correlation to assess monotonic relationships between variables; ii) Pillai's

trace test to analyze multiple dependent variables (Rn, H, LE and $FCO_2$) in relation to the independent variable (AOD); iii) Random Forest machine learning to determine aerosol importance in relation to flux variables, generating a model for each flux (Rn, H, LE and $FCO_2$) as a dependent variable and the variables in Table 1 as independent variables.

Pillai (1955) developed equation 5, which is widely used in analyses such as multivariate analysis of covariance (MANCOVA), particularly when assumptions are violated, such as when the covariance matrices are inhomogeneous.

$$p(T^{(s)}) = \frac{(T^{(s)})^{sm}}{(1+T^{(s)})^{s(2m+2n+s+1)/(2+1)}\beta[sm+1,\frac{1}{2}s(2n+s+1)]}, 0 < T^{(s)} < \infty \qquad (5)$$

Where, 'm' and 'n' are related to the sample sizes, 's' is a rank defined by the sum of the products of the matrix, 'p' are the characteristic roots of the matrix with values ranging from 0 to 1, 'T' summarizes the magnitude of the difference between the groups in relation to all dependent variables and '$\beta$' is used to calculate the p-value associated with the Pillai trace statistic.

A $p(T^{(s)})$ value close to zero indicates that there is greater overlap between categorical groups in a multivariate space. A

p-value below 5 % indicates that there is statistical significance between the dependent variables in relation to the categorical groups.

Finally, the importance of aerosols in relation to flux variables was determined using the Random Forest machine learning algorithm. According to Biau and Scornet (2016), this method provides accurate and reliable predictions because it combines multiple decision trees, thus minimizing the risk of overfitting — when the model fits the training data too well, compromising

its ability to generalize. The algorithm follows the 'divide and conquer' principle by sampling fractions of the data, building randomized tree predictors for each subset, and aggregating the results.

The estimation of the finite nonparametric regression forest presents the combination of decision trees by Equation 6 (Biau and Scornet, 2016).

$$\hat{m}_{M,n}(x;\Theta_1,\ldots,\Theta_M,D_n) = \frac{1}{M}\sum_{j=1}^{M} m_n(x;\Theta_j,D_n) \qquad (6)$$

Where: $\hat{m}$ is the prediction of the model for an observation $x$; M is the number of random regression trees; $n$ is the number of observations in the training set $D_n$; the values $\Theta_1,\ldots,\Theta_M$ are independent random variables associated with each tree $j$; and the predicted value at the query point $x$ is denoted by $m_n(x;\Theta_j,D_n)$.

Thus, the Random Forest is an ensemble method that combines several decision trees (M trees), where each tree ($j$) makes a prediction $m_n(x;\Theta_j,D_n)$ for observation $x$, arriving at an average of the predictions of all trees.



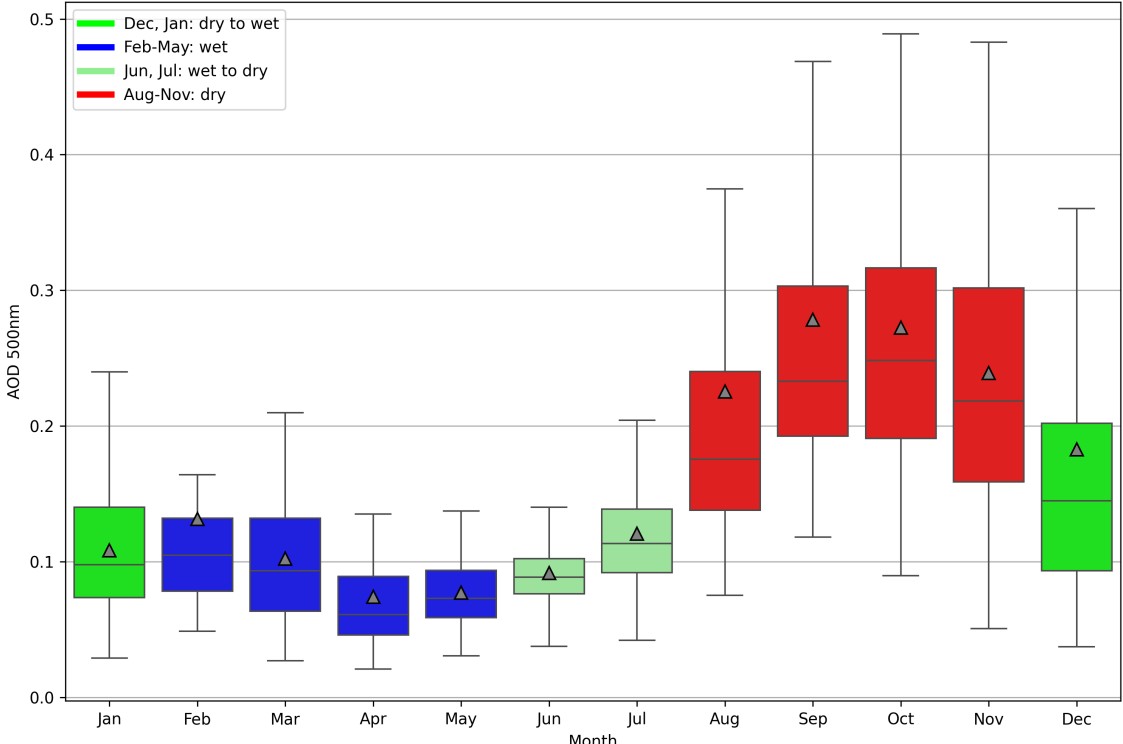

**Figure 2.** Box plot showing monthly AOD 500 nm values measured at the ATTO site between 2016 and 2022. The box represents the central 50 % of the data, the whiskers represent the smallest and largest non-outlier values, while the means are indicated by the green triangles and the medians are the lines inside the box.

## 3  Results and Discussion

### 3.1  Characteristics of seasonal aerosol variation

The distribution of atmospheric aerosols, expressed as AOD, exhibits a clear seasonal cycle at the ATTO site (Figure 2). The lowest values occur during the wet season, with an average of 0.07 in April, while the dry season is marked by higher AOD values, reaching an average of 0.28 in September. Furthermore, this seasonal variation in AOD values has previously been observed at other sites in the Amazon region (Artaxo et al., 2013; Cirino et al., 2014; Palácios et al., 2022). Cirino et al. (2014), for example, used data measured at the ZF2 site, located 60 km northwest of Manaus in central Amazonia, to show that AOD values were close to 0.4 (with peaks above 0.5) in the dry season and less than 0.2 in the wet season. Attention is drawn to the AOD values observed in the southern region of the Amazon basin, which is influenced by the arc of deforestation, an agricultural frontier zone with intense burning activity during the dry season (Davidson et al., 2012). Several studies in this region have shown that AOD values often exceed 4 in the dry season, whereas in the wet season they rarely exceed 0.2 (Fuzzi et al., 2007; Artaxo et al., 2013; Palácios et al., 2024).





**Table 2.** Averages of the radiation components in the period from 10:00 to 14:00 LT, during the dry season from 2016 to 2022, with the respective relative difference between the polluted and clean regimes.

| Means of radiation variables | | | |
|---|---|---|---|
| Variables | Polluted | Clean | Relative Difference |
| SWin ($\mathrm{W\,m^{-2}}$) | 797.61 ±129.33 | 844.81 ±145.27 | -5.59 |
| SWout ($\mathrm{W\,m^{-2}}$) | 94.79 ±14.67 | 94.94 ±16.70 | -0.16 |
| LWatm ($\mathrm{W\,m^{-2}}$) | 431.99 ±9.61 | 428.11 ±10.14 | 0.90 |
| LWterr ($\mathrm{W\,m^{-2}}$) | 483.61 ±8.95 | 487.67 ±11.32 | -0.83 |
| Rn($\mathrm{W\,m^{-2}}$) | 617.33 ±103.72 | 656.98 ±117.59 | -6.04 |

The main distinction between the AOD values measured above the ATTO site and those measured in the southern Amazon is the magnitude of these values. In particular, the AOD values at the ATTO site are approximately 15 times lower than those in the region close to the arc of deforestation during the dry season (Sena et al., 2013; Palácios et al., 2020). Andreae et al.
(2015), Pöhlker et al. (2018) and Holanda et al. (2023) for example, investigated the seasonal contrast of aerosols at the ATTO site, highlighting that parts of the wet season resemble preindustrial conditions with minimal human impact.

Figure 3 shows the average daily AOD values for the dry and wet seasons, from 2016 to 2022. It is clear to see that the highest average AOD values were obtained during the dry season, with values exceeding 1.5, while in the wet season these values did not exceed 0.5, a result similar to that already reported in Figure 2. It should also be noted that during the dry season,
the 90th and 10th percentiles of the AOD values are 0.40 and 0.13, respectively. During the wet season, these percentiles were 0.13 and 0.04, respectively. In other words, the AOD values above the 90th percentile in the wet season are slightly higher than the values observed for the 10th percentile in the dry season. This reinforces what was already mentioned in Figure 2, that the wet season in the ATTO region is quite 'clean' compared to the dry season. As the main goal of this work is to investigate the impact of the presence of aerosols on surface turbulent fluxes, from now on we will only work with data from the dry season,
and for this we will use two classes: i) "clean regime" being the one in which the AOD values were below the 10th percentile, and ii) "polluted regime" being the one in which the AOD values were above the 90th percentile.

### 3.2 Relationship between AOD and surface turbulent fluxes

The average values of the radiation balance components during the period of greatest radiative and convective activity between 10:00 and 14:00 LT in the ATTO are shown in Table 2. The negative sign in the difference between the polluted and clean
regimes indicates that the radiative components decrease during this period. The Rn fell the most in relative terms, by around -6 %. Reflected shortwave radiation (SWout) was the least affected by pollution, with a decrease of just -0.16 %. As is well known, the longwave balance is always negative during the daytime in the Amazon region (von Randow et al., 2004) since LWterr is greater than LWatm. However, pollution reduced the difference between LWatm and LWterr by around $8\,\mathrm{W\,m^{-2}}$ compared to the clean regime, indicating a slightly less radiative surface and a slightly warmer atmosphere.



**Figure 3.** (a) and (c) Daily AOD averages (500 nm), (b) and (d) their respective histograms. Values above the red line indicate high aerosol concentration (above the 90th percentile), while values below the blue line indicate low aerosol concentration (below the 10th percentile).

Quantifying the impact of aerosols on radiative flux remains a significant challenge in climate system studies, with persistent uncertainties (Palácios et al., 2022). However, the relationship between aerosols and radiative flux has been investigated for decades in the Amazon region (Ross et al., 1998; Procopio et al., 2004; Rizzo et al., 2011; Artaxo et al., 2013; Palácios et al., 2022).

There is a consensus in the literature that an increase in AOD reduces SWin, which consequently also causes a reduction in

Rn. However, the magnitude of these reductions varies considerably. Studies carried out during the dry season in the Amazon rainforest using different methods to estimate direct aerosol radiative forcing (ARF) illustrate this variability. For example, Ross et al. (1998) reported an average daily ARF of -20 $\pm$ 7 $\mathrm{W\,m^{-2}}$ per unit of AOD at 550 nm in the Amazon rainforest. In contrast, Palácios et al. (2022) estimated an average ARF of -20.77 $\pm$ 5.04 $\mathrm{W\,m^{-2}}$ for the dry season in the central Amazon.



Procopio et al. (2004) found daily ARF values (ARF24h) ranging from -21 to $-74\,\mathrm{W\,m^{-2}}$ in the deforestation arc, an area
with higher levels of pollution than the central Amazon. Rizzo et al. (2011) investigated this central region and reported an
ARF24h value of $-32\,\mathrm{W\,m^{-2}}$.

Although these studies provide estimates of the reduction in surface radiation from aerosols in the Amazon, they do not
converge on a single consensus value. This is because, in addition to the different methodologies used to obtain ARF values,
Procopio et al. (2004), Sena et al. (2013) and Palácios et al. (2020, 2022) point out that uncertainties lie mainly in the complex
interactions between types and concentrations of aerosols, surface characteristics, atmospheric conditions, and solar geometry.

SWout is directly related to surface albedo and the fact that it did not change significantly in our data between regimes
(maintaining albedo at $\sim 0.11$) indicates that pollution has a secondary effect compared to the characteristics of the surface
itself. There is a wide range of surface characteristics in the Amazon that directly influence albedo, as observed by von Randow
et al. (2004) and Pareja-Quispe et al. (2021): i) degree of vegetation cover; ii) soil and vegetation water conditions; iii) solar
elevation; iv) cloud cover and; v) wind speed and direction.

However, the behavior of longwave radiation was quite interesting. It shows that because of their interaction with the incident
shortwaves, aerosols increase the emission of thermal energy toward the surface. At the same time, they act as a barrier to the
total energy reaching the surface, thus impacting the amount of thermal energy emitted by the surface itself. The increase in
LWatm and the decrease in LWterr in the polluted regime result in a smaller longwave balance in this regime. de Menezes Neto
et al. (2016) also observed this effect in their experiments involving biomass burning aerosols in South America: a subtle
variation in longwave intensity attributed to the presence of aerosols.

With reduced solar energy input on the surface during the polluted regime, cooling occurs at the forest-atmosphere interface,
accompanied by a decrease in VPD compared to the clean regime, as illustrated in Figure 4. The cooling between the 10:00
and 14:00 LT regimes implies an average temperature reduction on the canopy surface of 0.53° C, resulting in a -3.21 hPa (19.5
%) in VPD. As the curve for the clean regime is consistently above that for the polluted regime at all shown temperatures, it is
suggested that the clean regime will first achieve a reduction in evapotranspiration, given the approximately linear relationship
between temperature and VPD.

These cooling values are consistent with the effects documented in other studies. For example, Moreira et al. (2017) found
a reduction in 1.2° C above the Amazon region, while Cirino et al. (2014) identified a 1.8° C and a decrease in 35 % in VPD
in the central Amazon. In the deforestation arc, Rodrigues et al. (2024) found an average cooling effect of between 3° C and
4° C, as well as reductions of between -2 and -3 hPa in VPD.

Braghiere et al. (2020) investigated temperature variations in the Amazon using a radiative transfer model. By simulating a
scenario without aerosols (AOD = 0) and comparing it with real conditions, they observed an increase in temperature in the
scenario without aerosols. They identified a correlation between relative irradiance, air temperature, and VPD. Meanwhile,
Herbert and Stier (2023) and Palácios et al. (2024) reinforce the idea that AOD significantly influences temperature variations,
particularly on a regional scale.

Herbert and Stier (2023) and Palácios et al. (2024) also highlight that the physical characteristics of the aerosols present in
the atmosphere, such as size, mixing state and presence of coatings, as well as the chemical characteristics, such as the ability to





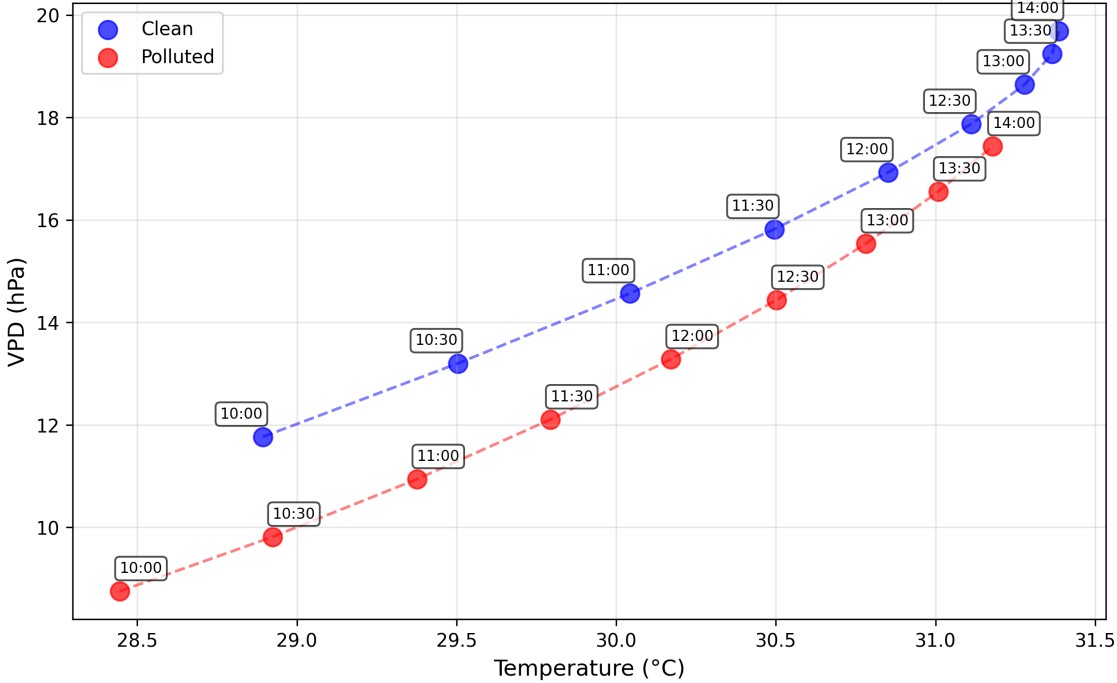

**Figure 4.** Relationship between temperature and VPD above the forest canopy at the ATTO for the different regimes (clean and polluted) during the dry season from 2016 to 2022.

absorb or scatter light and hygroscopicity, determine their direct impact on temperature and VPD through radiative interaction, as well as their indirect impact by influencing cloud properties and evapotranspiration rates. These are essential components of the atmosphere's energy balance.

The interaction between aerosols, radiation, and evapotranspiration affects not only temperature and VPD, but also the fluxes of energy and matter on the surface. This has a direct impact on atmospheric and ecosystem processes. Figure 5 illustrates the impact of aerosols on these fluxes. It can clearly be seen that for the polluted regime, the values were lower than those observed during the clean regime, especially during periods of high solar radiation, i.e. between 10:00 and 14:00 LT. As previously mentioned, the most significant reductions in the energy available to the surface occur during this period, with Rn falling by -6 %, as reflected in the energy partitions.

Sensible heat decreased by an average of $-17.30\,\mathrm{W\,m^{-2}}$ (11.3 %), reflecting reduced energy transfer to the atmospheric boundary layer. Similarly, LE decreased by $-45.08\,\mathrm{W\,m^{-2}}$ (10.4 %), indicating limited evapotranspiration due to the reduced radiative energy available. The Bowen ratio, which relates H and LE, remained at 0.35 for both regimes, suggesting that a higher proportion of energy was allocated to latent processes, as expected in forest environments. The sum of H and LE was also found to be $67.85\,\mathrm{W\,m^{-2}}$ lower for the clean regime than for Rn, while for the polluted regime, this value was





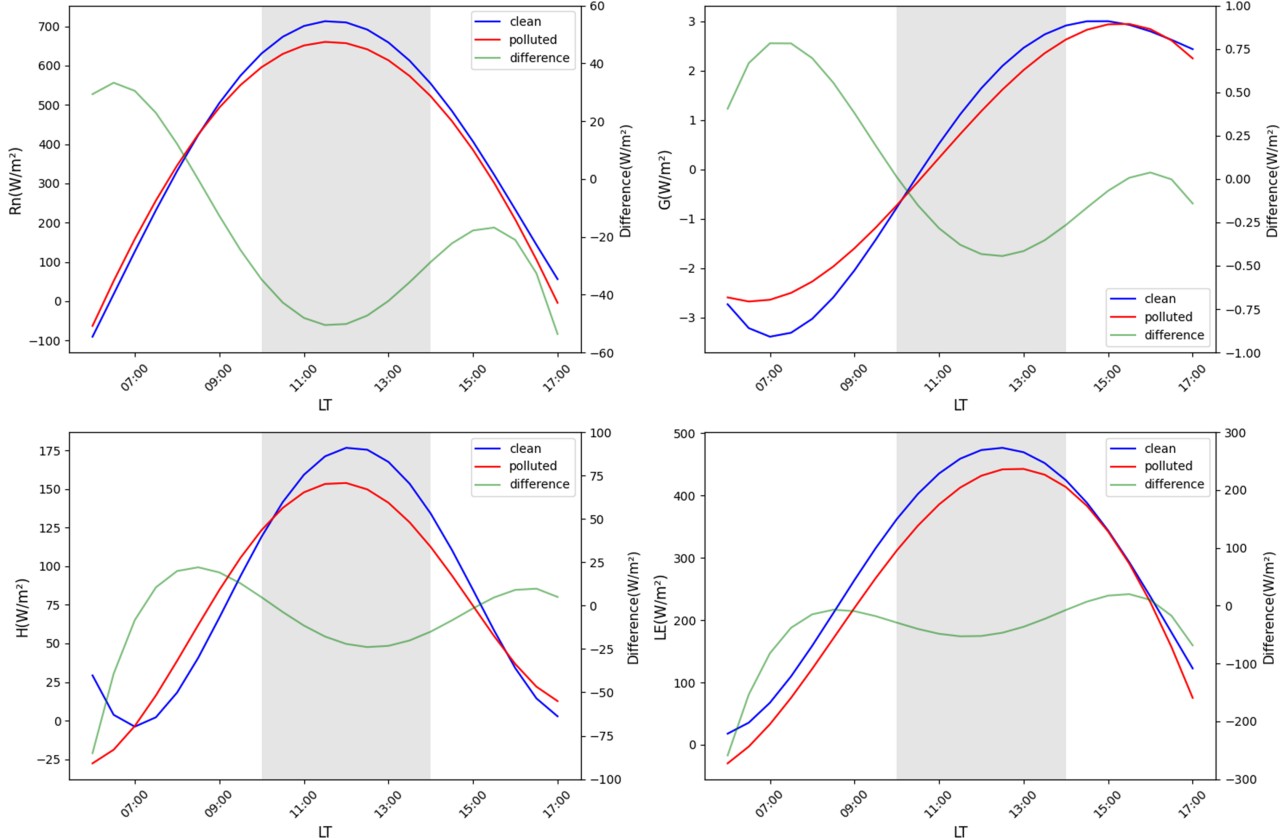

**Figure 5.** Diurnal cycle of surface fluxes in the dry season (2016-2022), with emphasis on the time between 10:00 and 14:00 LT. In blue are the curves with data from days in the clean regime, in red the data from days in the polluted regime and in green the respective difference between the variables.

$91.18\,\mathrm{W\,m^{-2}}$. It appears that the polluted regime is further from the energy balance close, suggesting a change in how this energy is distributed.

The ground heat flux (G) also decreased by $-0.60\,\mathrm{W\,m^{-2}}$ (39.3 %), demonstrating its greater sensitivity to variations in Rn compared to turbulent fluxes.

     In addition to their effect on energy fluxes, aerosols were found to have a significant influence on flux $CO_2$, showing an average drop of $-5.69\,\mathrm{\mu mol\,m^{-2}\,s^{-1}}$ (57.7 %) in the polluted regime between 10:00 and 14:00 LT. This is when the difference between the polluted and clean regimes is most pronounced, indicating that the forest absorbs more $CO_2$ in the polluted regime

(Figure 6).

     In the polluted regime, an increase in gas exchange due to photosynthesis is visible in Figure 6, suggesting a possible increase in evapotranspiration. However, analysis of the energy available for evapotranspiration (LE) shows a consistent decrease in the polluted regime compared to the clean regime (Figure 5), which contradicts this expectation.





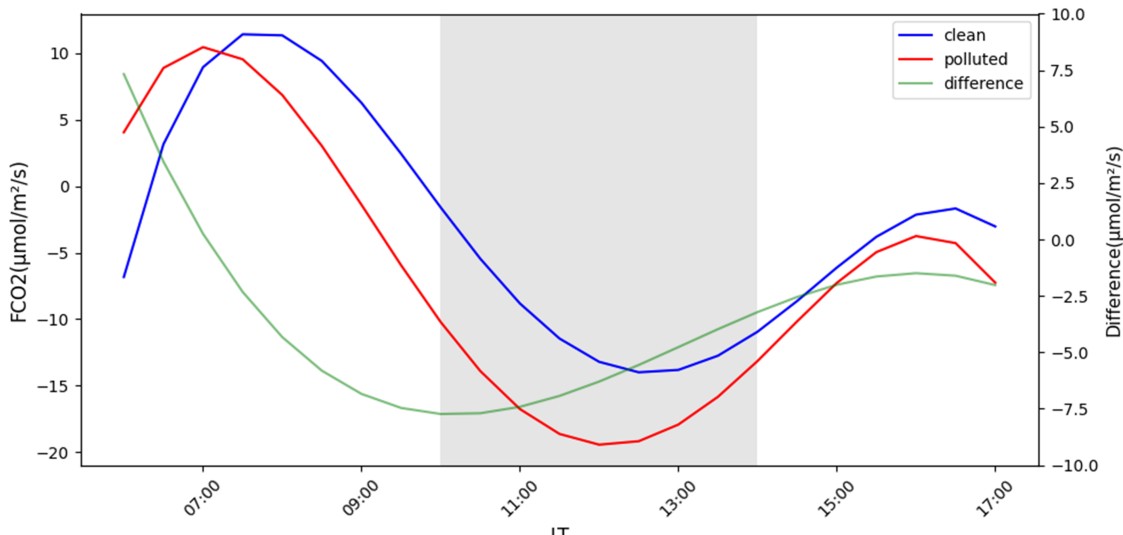

**Figure 6.** The diurnal cycle of matter flux in the ATTO region between 06:00 and 17:00 LT, fitted with a fourth-order polynomial. The data are from the dry season between 2016 and 2022, with emphasis on the period between 10:00 and 14:00 LT. The blue curves show the data from days in the clean regime, the red curves show the data from days in the polluted regime, and the green curves show the difference between the two.

The apparent paradox of an increase in $CO_2$ absorption alongside a reduction in LE can be explained by water use efficiency

(WUE). According to Dekker et al. (2016) and Yang et al. (2016), WUE is defined as the ratio of carbon assimilated to water transpired by vegetation. Studies in the Amazon indicate that higher temperatures and consequently higher VPD generally result in lower WUE (Aguilos et al., 2018; Botía et al., 2021). Our results (Figure 4) show that, at peak solar radiation (12:30 LT), temperatures of 30.5° C and 31.2° C and VPD values of 14.2 and 18.0 were observed in polluted and clean regimes, respectively. These values suggest a higher efficiency in carbon assimilation in the polluted regime, as shown in Figure 6, and

less water is emitted by evapotranspiration, as shown in Figure 5, with fewer LE assigned to this process.

In forests in the USA, Steiner et al. (2013) conducted experiments to quantify the impact of aerosols on turbulent surface fluxes, observing reductions in H and LE ranging from 10 % to 30 %. Few studies have examined the relationship between H, LE and AOD in the Amazon region. Zhang et al. (2008), for example, used regional modeling with an AOD threshold of 0.3 to obtain a daily average reduction of $-15\,\mathrm{W\,m^{-2}}$ for H and $-5\,\mathrm{W\,m^{-2}}$ for LE. In the deforestation zone, Braghiere et al. (2020)

observed a decrease of $-67\,\mathrm{W\,m^{-2}}$ (36 %) for H and $-4\,\mathrm{W\,m^{-2}}$ (2 %) for LE when simulating clean conditions (AOD = 0) and comparing them with real conditions involving the presence of aerosols.

These results suggest that regional climate models may underestimate the reduction in LE, highlighting the importance of biological processes, such as transpiration, in compensating for these effects.

In contrast, numerous studies in the Amazon have demonstrated the significant impact of aerosols on $CO_2$ assimilation

by forests. This occurs by increasing the diffuse fraction of photosynthetically active radiation reaching forest shade zones,





**Table 3.** Influence of aerosols on surface turbulent fluxes, considering all 370 runs of data (see section 2.2), from 06:00 to 17:00 LT, at 30-minute average intervals. The MSE of H and FCO was obtained by cross-validation, a technique applied to improve the accuracy of the model due to less adjusted initial values. The MSE for Rn and LE did not require cross-validation, as they already showed an adequate fit.

| The influence of aerosols on energy and matter fluxes | | | |
|---|---|---|---|
| Model | AOD | MSE | $R^2$ |
| Rn | 0.027 | 0.19 | 0.81 |
| H | 0.030 | 0.30 | 0.65 |
| LE | 0.032 | 0.28 | 0.74 |
| $FCO_2$ | 0.066 | 0.38 | 0.46 |

thereby intensifying photosynthesis. Simultaneously, it reduces the net direct solar radiation reaching the canopy surface, thereby generating photosynthetic enhancement in this region (Doughty et al., 2010; Cirino et al., 2014; Rap et al., 2015; Moreira et al., 2017; Malavelle et al., 2019; Rodrigues et al., 2024). This diffuse fraction, which falls within the wavelengths of interest for vegetation (0.4 to 0.7 $\mu$m), can increase from around 19 % (the typical value of a clean atmosphere) to 80 %
under biomass burning conditions (Yamasoe et al., 2006).

### 3.3 Influence of aerosols on surface turbulent fluxes

The statistical correlations between aerosols and surface turbulent fluxes reveal monotonic relationships of low intensity or with no statistical significance. Furthermore, MANCOVA analysis with Pillai trace indicated a highly significant multivariate effect (p < 0.001). These findings suggest that the impact of aerosols on fluxes cannot be fully explained by linear relationships
alone, but rather by nonlinear interactions with other environmental factors such as temperature, humidity and wind.

Table 3 shows the relative contribution of AOD to the prediction of each flux model value, as obtained using the Random Forest machine learning (RFM) technique and the variables in Table 1. This was done to capture the nonlinear influence between aerosols and surface turbulent fluxes.

The results of the RFM (see Table 3) suggest that Rn performed best in terms of predictive capacity, with an $R^2$ of 0.81 and
an MSE of 0.19. In contrast, $FCO_2$ performed the worst, with an $R^2$ of 0.46 and an MSE of 0.38. The lower value $R^2$ for $FCO_2$ can be attributed to the greater complexity of the interactions between aerosols and the biophysical processes that regulate this flux. However, AOD had a significantly greater impact on the predictive performance of $FCO_2$ (6.6 %), which is more than double the values observed in the energy fluxes (LE: 3.2 %; H: 3.0 %; Rn: 2.7 %). Rn showed the least influence of all the energy fluxes.

These results corroborate the analyses in Section 3.2, which examined the period of the greatest radiative and convective activity and identified significant reductions in fluxes under the influence of AOD. The most significant reduction was recorded for $FCO_2$ (-57.7 %), followed by a balance between H (-11.3 %) and LE (-10.4 %), while the smallest reduction was observed for Rn (-6 %).



In summary, these findings demonstrate that aerosols influence surface fluxes in a complex, nonlinear manner, with varying
impacts depending on the type of flux analyzed. FCO$_2$ shows greater sensitivity, reflecting intricate biophysical processes,
while energy fluxes respond more moderately, being more closely linked to the impact of these aerosols on energy partitioning.

## 4   Conclusions

This study assessed, for the first time, the impact of aerosol regimes on the exchange of surface energy (net radiation - Rn,
sensible heat - H and latent heat - LE) and mass (carbon dioxide flux - FCO$_2$) at the forest-atmosphere interface in the central
Amazon, a region that experiences relatively pristine atmospheric conditions during part of the year. Based on long-term data
collected between 2016 and 2022 at the ATTO site, our analysis provides clear and quantitative evidence that high aerosol
loads (AOD > 0.40) reduced the magnitude of FCO$_2$, H, and LE fluxes compared to clean conditions (AOD < 0.13).

During the peak radiation period (10:00-14:00 LT), the polluted regime (AOD > 0.40) substantially reduces turbulent energy
fluxes, decreasing H by $17.30\,\mathrm{W\,m^{-2}}$ (11.3%) and LE by $45.08\,\mathrm{W\,m^{-2}}$ (10.4%). Simultaneously, the forest's net CO absorp-
tion increased, with FCO$_2$ decreasing by $-5.69\,\mathrm{\mu mol\,m^{-2}\,s^{-1}}$ (57.7%), indicating a significant increase in carbon assimilation.
This biophysical response was accompanied by a cooling of the forest-atmosphere interface by 0.53°C and a reduction in the
vapor pressure deficit (VPD) by 3.21 hPa (19.5%). Thus, aerosols also play an important role in modulating energy partitioning
and gross primary productivity in the tropical forest ecosystem.

Our findings indicate that even in the relatively pristine central Amazon during the dry season, a threshold aerosol load
(AOD   0.40) exists, above which significant impacts on energy fluxes occur. This suggests that in regions with higher aerosol
loads, such as the southern Amazon's arc of deforestation, impacts on energy balance could be even more severe.

Our statistical analyses indicate that aerosols and surface turbulent fluxes interactions are predominantly indirect and non-
linear, mediated by environmental variables like radiation, temperature, and humidity. Consequently, different inflection points
likely exist across the Amazon, and the AOD threshold identified here cannot be applied to the entire region. Furthermore, iso-
lating the aerosol effect from clouds requires rigorous filtering and a significant data collection effort, as cloud-free moments
are scarce in long-term Amazonian time series.

Our work advances knowledge by quantifying the simultaneous effects of aerossol on energy and matter fluxes, bringing
with it possibilities for improvements in climate models for the Amazon region and opening up the possibility of future work
aimed at coupling the carbon and water cycles, mediated by aerosols, shedding light on the functioning of forest ecosystems.
All of this is possible with the integrated analysis of diffuse radiation and the efficient use of water combined with the impact
of aerosols on energy and matter fluxes.

In addition, future work involving remote sensing and data from micrometeorological towers throughout the Amazon is
crucial in order to spatialize the results of all these dynamics between the forest-atmosphere interface, which is essential for
quantifying the impact of aerosols on the Amazonian climate system.



*Author contributions.* Conceptualization: MABdR, CQDJ, JCPCand FAFDO. Data curation: CQDJ, ACdA, CP, SR and PA. Formal analysis: MABdR, CQDJ. Funding acquisition: CQDJ. Investigation: MABdR, CQDJ and FAFDO. Methodology: MABdR, CQDJ, FAFDO and RSP. Project administration: CAQ, CQDJ. Resources: CQDJ, ACdA, CP, SR and PA. Software: MABdR and FAFDO. Supervision: CQDJ and RSP. Validation: MABdR and FAFDO. Writing (original draft preparation): MABdR, CQDJ. Writing (review and editing): MABdR, CQDJ, JCPC, FAFDO, ACSM, CP, SR, ACdA, MAF, PA, CAQ, RSP.

*Competing interests.* The authors declare that they have no conflict of interest.

*Acknowledgements.* Mariano A.B. da Rocha thanks the Coordenação de Aperfeiçoamento de Pessoal de Nível Superior - Brasil (CAPES) for the PhD grant awarded through the Environmental Science Graduate Program (PPGCA/UFPA). Cléo Q. Dias-Júnior acknowledges support from CNPQ (Processes 440170/2022-2, 406884/2022-6, 307530/2022-1). This study is part of the Amazon Tall Tower Observatory (ATTO), funded by the German Federal Ministry of Education and Research (BMBF, contracts 01LB1001A and 01LK1602A), the Brazilian
Ministry of Science, Technology and Innovation (MCTI/FINEP, contract 01.11.01248.00) and the Max Planck Society (MPG). ATTO is also supported by the Fundação de Amparo à Pesquisa do Estado do Amazonas (FAPEAM), Fundação de Amparo à Pesquisa do Estado de São Paulo (FAPESP), Universidade do Estado do Amazonas (UEA), Instituto Nacional de Pesquisas Amazônia (INPA), Programa de Grande Escala da Biosfera-Atmosfera na Amazônia (LBA) and the SDS/CEUC/RDS-Uatumã.



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
