# Peer review of "Observed Impacts of Aerosol Regimes on Energy and Carbon Fluxes in the Amazon Forest"

_EGUsphere, 2025_

## Author Comment (AC2)

**Title: Observed Impacts of Aerosol Regimes on Energy and Carbon Fluxes in the Amazon Forest**

Response (blue color) to anonymous Referee #1 (black). The original manuscript was changed accordingly. The lines indicated in our answers correspond to the track version of the manuscript.

**General comment**

The manuscript investigates how contrasting aerosol optical depth (AOD) regimes affect surface energy and carbon fluxes over an undisturbed Amazon rainforest using long-term in situ data (2016–2022) from the Amazon Tall Tower Observatory (ATTO). The authors focus on differences between "clean" (AOD < 0.13) and "polluted" (AOD > 0.40) regimes and assess impacts on radiation, sensible and latent heat fluxes, and $CO_2$ exchange. The topic is highly relevant to ACP because it addresses aerosol–biosphere–atmosphere interactions in one of the planet's key ecosystems. The study provides new observational insights from a unique long-term dataset and uses appropriate statistical tools (Spearman correlation, Pillai's trace, Random Forest) to assess nonlinear relationships. I think the paper is well written and it is neatly exposed. The literature cited is adequate.

We would like to thank Referee #1 for their detailed review of our manuscript and for the positive feedback. We are also grateful for the valuable contributions that helped to clarify the text and refine the analyses.

The manuscript presents an interesting empirical analysis of aerosol effects on energy and carbon fluxes in the Amazon. However, the methodology lacks quantitative robustness in **(i)** defining aerosol pollution regimes and in assessing statistical significance of differences between them and improved discussion. The structure and figures are generally clear, but the **(ii)** discussion often repeats background concepts and lacks a mechanistic synthesis connecting radiation, energy partitioning, and ecosystem carbon exchange.

We thank the Reviewer #1 for these important comments. They will certainly help to improve the methodology and discussion of the results.

We would like to begin our responses by stating that in the new version of the manuscript, we regrouped our data in a way that allowed us to include a greater number of runs (half-hour periods).  In the previous version of the manuscript, in addition to excluding all periods when clouds were present, which is very common in the Amazon, we also excluded all data from a given day and time when a variable was missing. For example, if we did not have the reflected shortwave radiation measurement for a given time, we removed all other variables for that same time. This resulted in only 523 valid half-hour periods (370 dry season, 153 wet). In the new version of the Manuscript, we decided to regroup the variables so that they did not depend on each other. We first identified the periods in which we had the Clean and Polluted regimes (AOD < 0.13 and AOD > 0.40) and then identified how many runs of each variable were available for each regime. After this procedure, the number of runs
increased substantially, as shown in Table R1, comparison between the dataset used
in the first version of the manuscript (single database) and the dataset used for this
new version (database by variable).

Table R1: Number of runs (half-hour periods) after all quality controls mentioned in section 2.2.

| Variables | Single database | | | | | Database by variable | | | | |
|---|---|---|---|---|---|---|---|---|---|---|
| | 10:00 -14:00 LT | | 07:00 -17:00 LT | | Total | 10:00 -14:00 LT | | 07:00 -17:00 LT | | Total |
| | No. Clean | No. Polluted | No. Clean | No. Polluted | No. Sample | No. Clean | No. Polluted | No. Clean | No. Polluted | No. Sample |
| $SW_{in}(Wm^{-2})$ | 98 | 81 | 219 | 151 | 370 | 301 | 204 | 736 | 459 | 1195 |
| $SW_{out}(Wm^{-2})$ | 98 | 81 | 219 | 151 | 370 | 301 | 204 | 736 | 459 | 1195 |
| $LW_{atm}(Wm^{-2})$ | 98 | 81 | 219 | 151 | 370 | 301 | 200 | 733 | 453 | 1186 |
| $LW_{terr}(Wm^{-2})$ | 98 | 81 | 219 | 151 | 370 | 301 | 204 | 735 | 459 | 1194 |
| $Rn(Wm^{-2})$ | 98 | 81 | 219 | 151 | 370 | 301 | 200 | 733 | 453 | 1186 |
| $H(Wm^{-2})$ | 98 | 81 | 219 | 151 | 370 | 197 | 192 | 455 | 389 | 844 |
| $LE(Wm^{-2})$ | 98 | 81 | 219 | 151 | 370 | 183 | 180 | 447 | 386 | 833 |
| $FCO_2(\mu molm^{-2}s^{-1})$ | 98 | 81 | 219 | 151 | 370 | 247 | 195 | 596 | 405 | 1001 |
| $G(Wm^{-2})$ | 98 | 81 | 219 | 151 | 370 | 301 | 218 | 741 | 487 | 1228 |

In the new version of the manuscript, we rewrote L107:

"*After filtering, the resulting dataset is summarized in Table S1 and S2.*"

**(i)** In the previous version of the manuscript the classification into "Clean" and
"Polluted" regimes was based on the 10th and 90th percentiles of the AOD distribution
at ATTO. However, following the reviewer's recommendation, we performed a
sensitivity analysis by applying the statistical test using three percentile thresholds
(10th/90th, 15th/85th, and 20th/80th) to quantitatively assess the robustness of the
regime separation (Table R2). The results show that the 10th/90th and 15th/85th
percentile thresholds provide stronger physical contrasts between aerosol regimes,
expressed as larger differences in median values of key variables, and are associated
with statistically significant differences. Conversely, the 20th/80th threshold leads to a
loss of statistical significance for several variables, indicating a dilution of the physical
contrast between regimes. Based on these tests, the threshold was maintained to the
10th/90th percentiles, as this choice preserves physically meaningful differences
between aerosol regimes.

Table R2. Results of the statistical test comparing all analyzed variables between clean and polluted
aerosol regimes for three AOD thresholds. Panels correspond to: 10th–90th percentiles (top), 15th–85th
percentiles (middle), and 20th–80th percentiles (bottom).

**Main Stage - Percentil 10 (AOD≤0.13), Percentil 90 (AOD≥0.40)**

| Variable | N Clean | Mean Clean | SD Clean | N Polluted | Mean Polluted | SD Polluted | U Statistic | p-value | Significance |
|---|---|---|---|---|---|---|---|---|---|
| SWin($Wm^{-2}$) | 301 | 836.5 | 165.2 | 204 | 813.5 | 124.4 | 35391 | 0.004 | ** |
| SWout($Wm^{-2}$) | 301 | 92.8 | 19.7 | 204 | 95.9 | 15.1 | 27859 | 0.077 | ns |
| LWatm($Wm^{-2}$) | 301 | 431.5 | 10.4 | 200 | 432.1 | 9.4 | 29439 | 0.677 | ns |
| LWterr($Wm^{-2}$) | 301 | 484.7 | 14.0 | 204 | 483.6 | 10.8 | 34148 | 0.032 | * |
| Rn($Wm^{-2}$) | 301 | 659.3 | 137.8 | 200 | 632.8 | 100.8 | 35671 | 0.000 | *** |
| H($Wm^{-2}$) | 197 | 160.6 | 67.8 | 192 | 138.9 | 61.4 | 22611 | 0.001 | *** |
| LE($Wm^{-2}$) | 183 | 426.7 | 136.8 | 180 | 417.8 | 146.7 | 17246 | 0.438 | ns |
| $FCO_2$($\mu molm^{-2}s^{-1}$) | 247 | -12.5 | 8.0 | 195 | -17.4 | 8.6 | 32125 | 0.000 | *** |
| G($Wm^{-2}$) | 301 | 1.8 | 1.6 | 218 | 0.8 | 1.4 | 43719 | 0.000 | *** |

**Wide Stage - Percentil 15 (AOD≤0.14), Percentil 85 (AOD≥0.36)**

| Variable | N Clean | Mean Clean | SD Clean | N Polluted | Mean Polluted | SD Polluted | U Statistic | p-value | Significance |
|---|---|---|---|---|---|---|---|---|---|
| SWin($Wm^{-2}$) | 423 | 829.1 | 166.2 | 272 | 816.6 | 131.5 | 62510 | 0.054 | ns |
| SWout($Wm^{-2}$) | 423 | 91.8 | 20.1 | 272 | 95.7 | 15.6 | 50534 | 0.007 | ** |
| LWatm($Wm^{-2}$) | 423 | 432.7 | 10.0 | 268 | 433.2 | 9.4 | 55655 | 0.688 | ns |
| LWterr($Wm^{-2}$) | 423 | 484.1 | 14.5 | 272 | 484.5 | 11.0 | 59785 | 0.382 | ns |
| Rn($Wm^{-2}$) | 423 | 654.7 | 138.0 | 268 | 634.7 | 107.7 | 64197 | 0.003 | ** |
| H($Wm^{-2}$) | 281 | 157.5 | 70.0 | 258 | 139.0 | 62.7 | 42045 | 0.001 | ** |
| LE($Wm^{-2}$) | 265 | 422.3 | 133.9 | 241 | 419.8 | 145.7 | 32675 | 0.651 | ns |
| $FCO_2$($\mu molm^{-2}s^{-1}$) | 354 | -12.6 | 8.2 | 263 | -16.7 | 8.5 | 60056 | 0.000 | *** |
| G($Wm^{-2}$) | 427 | 1.8 | 1.6 | 298 | 0.9 | 1.4 | 83989 | 0.000 | *** |

**Alternative Stage - Percentil 20 (AOD≤0.16), Percentil 80 (AOD≥0.32)**

| Variable | N Clean | Mean Clean | SD Clean | N Polluted | Mean Polluted | SD Polluted | U Statistic | p-value | Significance |
|---|---|---|---|---|---|---|---|---|---|
| SWin($Wm^{-2}$) | 547 | 822.9 | 172.0 | 372 | 828.2 | 132.4 | 103274 | 0.698 | ns |
| SWout($Wm^{-2}$) | 547 | 91.1 | 20.4 | 372 | 96.3 | 15.4 | 85935 | 0.000 | *** |
| LWatm($Wm^{-2}$) | 544 | 433.4 | 9.8 | 366 | 433.6 | 9.5 | 98174 | 0.723 | ns |
| LWterr($Wm^{-2}$) | 547 | 483.7 | 14.6 | 372 | 485.2 | 11.0 | 100283 | 0.712 | ns |
| Rn($Wm^{-2}$) | 544 | 650.5 | 144.8 | 366 | 644.3 | 112.1 | 105788 | 0.109 | ns |
| H($Wm^{-2}$) | 377 | 155.9 | 71.0 | 350 | 140.8 | 64.5 | 74993 | 0.001 | ** |
| LE($Wm^{-2}$) | 355 | 416.4 | 135.7 | 331 | 420.4 | 148.0 | 58402 | 0.893 | ns |
| $FCO_2$($\mu molm^{-2}s^{-1}$) | 472 | -12.7 | 8.3 | 364 | -16.0 | 8.5 | 105109 | 0.000 | *** |
| G($Wm^{-2}$) | 567 | 1.9 | 1.6 | 411 | 1.0 | 1.5 | 150612 | 0.000 | *** |

We clarified this point in the revised manuscript as follows:

Section 2.3:

L134-137: "*Daily averages of AOD values were obtained to investigate seasonal variability. Our analysis distinguishes two contrasting atmospheric conditions at the ATTO site, defined as "Clean" and "Polluted" using AOD thresholds derived from the dry-season AOD distribution. The Clean and Polluted regimes correspond to the 10th (AOD ≤ 0.13) and 90th (AOD ≥ 0.40) percentiles, respectively. Further details on the seasonal aerosol analysis are provided in Section 3.1 and Table S3.*"

Section 3.1:

L201-207: "*As the main goal of this work is to investigate the impact of aerosols on surface turbulent fluxes, the analysis focuses on data from the dry season. In addition, during the dry season there is more aerosol data since the could interference is much*

*less pronounced than during the wet season. Two aerosol regimes were defined based on percentile thresholds of the dry-season AOD distribution. Several percentile combinations were tested to assess the robustness of the regime separation. Based on this analysis, the 10th and 90th percentiles were selected to define the Clean (AOD ≤ 0.13) and Polluted (AOD ≥ 0.40) regimes, respectively, as they preserve physically meaningful differences between aerosol regimes (See table S1).*"

**(ii)** In the new version of the manuscript, we improved the methodology by adding a statistical analysis for the definition of Clean and Polluted regimes. We also improved the discussion of the results to avoid repeating background concepts.

In its present form, I recommend **major revisions** according to my specific comments before the manuscript can be considered for publication in *ACP*.

**Specific comments**

The study contributes observational evidence from a rare, pristine tropical forest site. The long-term dataset and the combination of aerosol and flux measurements are strengths. Nevertheless, the novelty is moderate, as the main conclusions - reduction of net radiation and turbulent fluxes under high AOD, accompanied by enhanced $CO_2$ assimilation - are qualitatively consistent with previous literature (e.g., Cirino et al. 2014; Braghiere et al. 2020; Palácios et al. 2022). **(i)** The novelty would be strengthened by including a quantitative analysis of diffuse versus direct radiation, or by exploring seasonally resolved patterns rather than aggregating all data into two AOD categories. **(ii)** Defining "Clean" (AOD < 0.13) and "Polluted" (AOD > 0.40) purely from percentiles is arbitrary. Include a sensitivity test or physical rationale for these cutoffs. **(iii)** To increase the scientific value of the study, the authors should demonstrate, through appropriate statistical testing, whether the observed reductions (≈10%) are robust across years and not driven by interannual variability.

**(i)** We thank the reviewer for their valuable comment. Diffuse radiation (SWd) measurements are available only for the year 2021 for our experimental site. Prior to this period, SWd was not measured at the ATTO site, and in 2022 the sensor experienced technical issues that affected data quality. Nevertheless, to address the reviewer's suggestion, we quantified the diffuse radiation fraction (Fd = SWd/SWin) for the available period (2021) and compared Fd between Clean and Polluted aerosol regimes. Our results indicate higher Fd values under Polluted regime compared to Clean regime (Figure R1). Specifically between 10:00 and 14:00 LT, the mean Fd values were 0.43 and 0.27 for Polluted and Clean regime, respectively, indicating an absolute difference of 0.16 between the two regimes (p<0.05). This is consistent with enhanced scattering of solar radiation associated with increased aerosol loading (Giorgi et al., 2002; Seinfeld and Pandis, 2016; Ezhova et al., 2018). Moreover, daytime $CO_2$ fluxes showed a non-linear dependence on Fd, with net $CO_2$ uptake increasing up to an Fd threshold (≈0.6) and decreasing at higher Fd values (Figure R2). This behaviour was consistent with the response of NEE (net ecosystem exchange) for diffuse radiation reported by Deng et al. (2022) for four forest sites in China and aligns with the global-scale mechanisms proposed by Mercado et al. (2009).

These results provide observational support for the proposed mechanism linking
aerosol loading, radiation partitioning, and ecosystem carbon exchange.

[Figure]

Figure R1. Diffuse radiation fraction (Fd = SWd/SWin) under Clean and Polluted aerosol regimes during
2021.

[Figure]

Figure R2. The relationship between $CO_2$ flux and diffuse radiation fraction (Fd) during 2021 at ATTO
site.

The Methodology and Results sections have been updated in the new version of the
manuscript as follows:

L87-90: "*Additionally, diffuse shortwave radiation (SWd) was measured using a SPN1*
*Pyranometer (Delta-T Devices) installed at 75 m above ground level. However, SWd*
*data were available only for 2021, prior to this year, SWd was not measured at the*
*ATTO site, and data from 2022 were excluded due to technical issues with the sensor.*"

L328-337: "*We quantified the diffuse radiation fraction (Fd = SWd/SWin) for the available period (2021) and compared Fd between Clean and Polluted aerosol regimes. Our results indicate higher Fd values under the Polluted regime compared to Clean regime (Figure S1). Specifically for the 10:00 and 14:00 LT interval, the mean Fd values were 0.43 and 0.27 for Polluted and Clean regime, respectively, indicating an absolute difference of 0.16 between the two regimes (p<0.05). This is consistent with enhanced scattering of solar radiation associated with increased aerosol loading (Giorgi et al., 2002; Seinfeld and Pandis, 2016; Ezhova et al., 2018). Moreover, daytime $CO_2$ fluxes showed a non-linear dependence on Fd, with net $CO_2$ uptake increasing up to an Fd threshold (~0.6) and decreasing at higher Fd values (Figure S2). This behaviour was consistent with the response of net ecosystem exchange for diffuse radiation reported by Deng et al. (2022) for four forest sites in China, and aligns with the global-scale mechanisms proposed by Mercado et al. (2009). These results provide observational support for the proposed mechanism linking aerosol loading, radiation partitioning, and ecosystem carbon exchange.*"

**(ii)** We assessed the robustness of our results in relation to different regime thresholds as detailed in the General comment (L59-71 and Table R2 of this document).

**(iii)** We analyzed 2020 and 2022 separately due to higher data availability, while the remaining years (2016, 2017, 2018, 2019, and 2021), which had lower data availability, were grouped into a single category (Remaining). The pattern observed in the preliminary version of the manuscript (Figs. 5 and 6) was consistent across individual years (Figure R3). That is, under polluted conditions, reductions in SWin lead to a decrease in Rn, followed by decreases in H, LE, G, and $FCO_2$.

The following sentence has been added to the revised manuscript.

L293: "*The reductions in H, LE, G, and $FCO_2$ shown in Fig. 6 and Fig. 7 were also observed across individual years (see Fig. S3).*"

[Figure]

Figure R3. Box plots of a) incoming shortwave radiation (SWin), b) net radiation (Rn), c) sensible heat
flux (H), d) latent heat flux (LE), e) ground heat flux (G), f) $CO_2$ flux (FCO$_2$). All variables under clean
(blue) and polluted (red) aerosol regimes for 2020, 2022, and the remaining years (2016, 2017, 2018,
2019 and 2021), grouped due to limited data availability. Triangles indicate the mean value.
Line 94-97. Only 523 valid half-hour periods (370 dry season, 153 wet) are quite small
relative to the six-year period. The statistical representativeness and interannual
variability need further discussion.
We modified our methodology to obtain as much data as possible, as detailed in lines
L36-50 and Table R1 of this document.
Line 173-174. The section on radiative fluxes should include a graph of the full diurnal
cycle of SW, LW, and Rn to visually demonstrate the 10:00 - 14:00 LT maximum. This
would strengthen the rationale for focusing on that time window.
We thank Reviewer #1 for this valuable comment. As recommended, we have included
Figure R4 in Section 3.2 of the manuscript (Figure 4 in the new version of the
manuscript), which shows the full diurnal cycles of shortwave (SW), longwave (LW),
and net radiation (Rn), and highlights the maximum between 10:00 and 14:00 LT.

[Figure]

Figura R4. Diurnal cycles of radiative fluxes during the dry season from 2016 to 2022: (a) incoming
(SWin) and (b) outgoing (SWout) shortwave radiation, (c) incoming atmospheric (LWatm) and (d)
outgoing terrestrial (LWterr) longwave radiation, and (e) net radiation (Rn). Markers indicate observed
data, and solid lines represent fourth-order polynomial fits, with the corresponding R² and RMSE.
In the revised version of the manuscript, we have added Figure R4 and the following
paragraph to clarify the choice of the analysis period:

*L209-212 : "As described in Section 2.3, the comparisons between Clean and Polluted*
*regimes were restricted to the 10:00–14:00 LT period, corresponding to the maximum*
*net radiation. The full diurnal cycles of shortwave, longwave, and net radiation during*
*the dry season (2016–2022) show that the maximum values occur between 10:00 and*
*14:00 LT (Figure 4), supporting the choice of this time window for the subsequent*
*analyses*."

Line 203-204. The physical interpretation of the longwave radiation components
(LWatm and LWterr) is interesting, but it would benefit from quantitative support - for
instance, by including a vertical temperature profile or an estimate of surface and
atmospheric emissivity.

We used the Stefan–Boltzmann equation: $LW = \varepsilon\sigma T^4$, where LW is the longwave
radiation ($Wm^{-2}$), $\varepsilon$ is the emissivity, $\sigma$ is the Stefan–Boltzmann constant ($5.67 \times 10^{-8}$
$Wm^{-2}K^{-4}$) e T is the absolute temperature (K).

Using this relationship, separately for the Clean and Polluted regimes, we estimated
the surface emissivity ($\varepsilon_s$) from the measured outgoing longwave radiation (LWterr)
and the infrared surface temperature (Ts), and the atmospheric emissivity ($\varepsilon_a$) from
the measured incoming longwave radiation (LWatm) and air temperature (Ta). The
resulting emissivity values are shown in Table 3. The emissivities exhibit very similar
values under both regimes, suggesting that the differences in aerosol conditions were
not sufficient to affect the surface emissivity or the atmospheric emissivity.

Table R3: Mean values of incoming and outgoing longwave radiation (LWatm and LWterr), air
temperature (Ta), surface infrared temperature (Ts), and the corresponding atmospheric ($\varepsilon_a$) and
surface ($\varepsilon_s$) emissivities, under Clean and Polluted regimes for the 10:00 and 14:00 LT interval.

| | LWatm ($Wm^{-2}$) | LWterr ($Wm^{-2}$) | Ta (°C) | Ts (°C) | $\varepsilon_a$ | $\varepsilon_s$ |
|---|---|---|---|---|---|---|
| Clean | 431.5 | 484.74 | 30.3 | 32.6 | 0.898 | 0.978 |
| Polluted | 432.1 | 483.60 | 30.0 | 31.7 | 0.902 | 0.988 |

Line 227-232. The manuscript would benefit from a discussion of the energy balance
closure, specifically addressing the discrepancy between Rn and the sum of H, LE,
and G. Reporting the residuals for both clean and polluted regimes would provide a
clearer evaluation of data quality and potential systematic biases.

We thank the reviewer for this important comment. Following this suggestion, we
updated the manuscript by adding the following text:

L278-281: "*The surface energy balance closure was 0.89 for the Clean regime and*
*0.88 for the Polluted regime, comparable to values reported in the literature (Mauder*
*et al., 2024). The corresponding residuals were of similar magnitude (70 $Wm^{-2}$ for*
*Clean and 75 $Wm^{-2}$ for Polluted), indicating that the observed differences in energy*
*fluxes are not related to differences in energy balance closure.*"

[Figure]

Figura R5. Energy balance closure for Clean and Polluted regimes during the dry season from 2016 to
2022, considering the 10:00–14:00 local time.

Figure 6. The fourth-order polynomial fits to the diurnal cycles provide a useful visual
comparison, but the authors should complement them with statistical analyses to
confirm that the apparent differences between regimes are statistically significant.

We thank the reviewer for this comment. The statistical analyses comparing Clean and
Polluted aerosol regimes were performed for all analyzed variables, including those
shown in Figure 6. The results are presented in Table R1, as requested in the General
Comments.

Line 250-255. The connection between aerosol effects and water-use efficiency (WUE)
is largely speculative because WUE is not quantitatively evaluated in the manuscript.
The authors should consider calculating WUE (for example, as GPP/ET using $FCO_2$
and LE data) or presenting an appropriate proxy to substantiate this aspect of the
discussion.

We thank the reviewer for this important comment. Following this suggestion, we now
quantify water-use efficiency (WUE) using $|FCO_2|$ / LE as a proxy, and performed
statistical analyses comparing Clean and Polluted aerosol regimes (Table R4).

Table R4: Mean water-use efficiency (WUE) calculated as $|FCO_2|$ / LE ($\mu mol J^{-1}$), under polluted and
clean regimes for individual years (2020 and 2022), the remaining years (2016, 2017, 2018, 2019 and
2021) and all years combined.

| Year | AOD Regime | $|FCO_2|$ / LE ($\mu mol J^{-1}$) |
|------|-----------|------------------------------------|
| 2020 | Clean | 0.022 |
| 2020 | Polluted | 0.042 |
| 2022 | Clean | 0.031 |
| 2022 | Polluted | 0.044 |
| Remaining | Clean | 0.026 |
| Remaining | Polluted | 0.049 |
| All Years | Clean | 0.029 |
| All Years | Polluted | 0.042 |

In the new version of the Manuscript we added the follow sentence:

L304-307: "*In this study, WUE was estimated using $FCO_2/LE$ as a proxy. WUE was significantly higher under Polluted compared to Clean regime(mean values of 0.042 and 0.029, respectively, p < 0.05). This indicates that under Polluted regime, vegetation assimilates more carbon per unit of water lost, consistent with the observed reduction in latent heat flux (Figure 6) despite enhanced $CO_2$ uptake (Figure 7).*"

Line 242-245. It seems to me that there is some inconsistency throughout the manuscript regarding the sign convention of $CO_2$ flux. The authors should clearly state that $CO_2$ uptake by the ecosystem corresponds to a negative flux, while positive flux values indicate a $CO_2$ emission to the atmosphere. Accordingly, a "drop" or decrease in $FCO_2$ should represent reduced carbon uptake, not enhanced assimilation. In the Abstract, for example, Authors should clarify the meaning of "decrease in $CO_2$ fluxes by 58%" (does this mean more negative flux, i.e., greater uptake?). Clarifying this point is essential for avoiding misinterpretation of the results and ensuring consistency across figures, tables, and the discussion.

We thank the reviewer for pointing out this important issue. To avoid any ambiguity, we have clarified throughout the manuscript that negative $CO_2$ fluxes indicate net ecosystem uptake. The text has been revised as follows:

L290-292: "*In addition to their effect on energy fluxes, aerosols were found to have a significant influence on the $CO_2$ flux, becoming more negative by an average of 4.9 $\mu mol\ m^{-2}s^{-1}$ (39.5 %) in the polluted regime compared to clean conditions between 10:00 and 14:00 LT.*"

Abstract: "*We find that enhanced aerosol presence reduces both sensible heat flux and energy available for evapotranspiration by approximately 13.5 % and 2.1% respectively, while increasing $CO_2$ uptake (i.e., $CO_2$ flux becoming more negative) by about 39.5 %.*"

The figures are generally clear and well designed, but they would benefit from the inclusion of confidence intervals or error bars to convey the statistical variability of the data. Adding uncertainty information would allow readers to better assess the robustness of the observed differences between regimes and the reliability of the fitted curves.

We thank the reviewer for this suggestion. Figures 4, 5, and 6 (Fig. 5, 6, 7 in the new version of the manuscript) have been revised to include the observed data points underlying the averaged curves, allowing a direct visualization of the data variability.

**Minor comments**

All physical variables ($Rn$, $H$, $LE$, $FCO_2$, $AOD$, etc.) should be written in italics or formatted with the equation editor for consistency and readability.

The entire manuscript was revised to ensure consistent formatting of all physical variables. Thanks.

Throughout the manuscript, several acronyms are not explicitly defined (ARF24h, LWterr), which may affect readability. I recommend defining each acronym upon first use.

The entire manuscript was revised to ensure that all acronyms are explicitly defined at their first occurrence. Thanks.

Line 191. "ARF24h", did Authors refer to daily mean? It should be clarified

Yes, in the revised manuscript, we have removed the term *ARF24h* to avoid ambiguity as follows:

L227-230: "*Consistent with these findings, Palácios et al. (2022) estimated an average ARF of −20.77 ± 5.04 Wm$^{-2}$ for the dry season in the central Amazon. Procopio et al. (2004) found daily ARF values ranging from −21 to −74 Wm$^{-2}$ in the deforestation arc, an area with higher levels of pollution than the central Amazon. Rizzo et al. (2011) investigated this central region and reported a daily ARF value of −32 Wm$^{-2}$.*"

Line 187-188. The phrase "In contrast" seems used incorrectly; the studies cited do not contradict one another, showing similar ARF values (within the estimated errors). The Authors should revise wording.

We thank the reviewer for this comment. The term "*In contrast*" was replaced by "*Consistent with these findings*" as detailed in previous comment (L352-356 in this document).

Table3. Caption. FCO should be replaced by FCO2

The caption of Table 3 has been revised.

Line 299. As before, CO -> CO2

The text has been updated accordingly.

**References**

Aguilos, M., Stahl, C., Burban, B., Hérault, B., Courtois, E., Coste, S., Wagner, F., Ziegler, C., Takagi, K., and Bonal, D.: Inter-annual and Seasonal Variations in Ecosystem Transpiration and Water Use Efficiency in a Tropical Rainforest, Forests, 10, 14, https://doi.org/10.3390/f10010014, 2018.

Botía, S., Komiya, S., Marshall, J., Koch, T., Gałkowski, M., Lavric, J., Gomes-Alves, E., Walter, D., Fisch, G., Pinho, D. M., Nelson, B. W., Martins, G., Luijkx, I. T., Koren, G., Florentie, L., Carioca de Araújo, A., Sá, M., Andreae, M. O., Heimann, M., Peters, W., and Gerbig, C.: The CO2 record at the Amazon Tall Tower Observatory: A new opportunity to study processes on seasonal and inter-annual scales, Global Change Biology, 28, 588–611, https://doi.org/10.1111/gcb.15905, 2021.

Braghiere, R. K., Yamasoe, M. A., Évora do Rosário, N. M., Ribeiro da Rocha, H., de Souza Nogueira, J., and de Araújo, A. C.: Characterization of the radiative impact of aerosols on CO2 and energy fluxes in the Amazon deforestation arch using artificial neural networks, Atmospheric Chemistry and Physics, 20, 3439–3458, https://doi.org/10.5194/acp-20-3439-2020, 2020.

Cirino, G. G., Souza, R. A. F., Adams, D. K., and Artaxo, P.: The effect of atmospheric aerosol particles and clouds on net ecosystem exchange in the Amazon, Atmospheric Chemistry and Physics, 14, 6523–6543, https://doi.org/10.5194/acp-14-6523-2014, 2014.

Deng, X., Zhang, J., Che, Y., Zhou, L., Lu, T., and Han, T.: The Effect of Diffuse Radiation on Ecosystem Carbon Fluxes Across China From FLUXNET Forest Observations, Frontiers in Earth Science, 10, https://doi.org/10.3389/feart.2022.906408, 2022.

Ezhova, E., Ylivinkka, I., Kuusk, J., Komsaare, K., Vana, M., Krasnova, A., Noe, S., Arshinov, M., Belan, B., Park, S. B., Lavric, J. V., Heimann, M., Petäjä, T., Vesala, T., Mammarella, I., Kolari, P., Bäck, J., Rannik, U., Kerminen, V. M., and Kulmala, M.: Direct effect of aerosols on solar radiation and gross primary production in boreal and hemiboreal forests, Atmospheric Chemistry and Physics, 18, 17 863–17 881, https://doi.org/10.5194/acp-18-17863-2018, 2018.

Giorgi, F., Bi, X., and Qian, Y.: Direct radiative forcing and regional climatic effects of anthropogenic aerosols over East Asia: A regional coupled climate-chemistry/aerosol model study, Journal of Geophysical Research Atmospheres, 107, AAC 7–1–AAC 7–18,425 https://doi.org/10.1029/2001JD001066, 2002.

Mercado, L. M., Bellouin, N., Sitch, S., Boucher, O., Huntingford, C., Wild, M., and Cox, P. M.: Impact of changes in diffuse radiation on the global land carbon sink, Nature, 458, 1014–1017, https://doi.org/10.1038/nature07949, 2009.

Palácios, R. d. S., Artaxo, P., Cirino, G. G., Nakale, V., Morais, F. G., Rothmund, L. D., Biudes, M. S., Machado, N. G., Curado, L. F. A., Marques, J. B., and Nogueira, J. d. S.: Long-term measurements of aerosol optical properties and radiative forcing (2011-2017) over Central Amazonia, Atmósfera, 35, 143–163, https://doi.org/10.20937/atm.52892, 2022.

Procopio, A. S., Artaxo, P., Kaufman, Y. J., Remer, L. A., Schafer, J. S., and Holben, B. N.: Multiyear analysis of amazonian biomass burning smoke radiative forcing of climate, Geophysical Research Letters, 31, https://doi.org/10.1029/2003gl018646, 2004.

Rizzo, L. V., Correia, A. L., Artaxo, P., Procópio, A. S., and Andreae, M. O.: Spectral dependence of aerosol light absorption over the Amazon Basin, Atmospheric Chemistry and Physics, 11, 8899–8912, https://doi.org/10.5194/acp-11-8899-2011, 2011.

Seinfeld, J. H. and Pandis, S. N.: Atmospheric chemistry and physics: from air pollution to climate change., John Wiley Sons, 3rd edn., 2016.

---

## Author Comment (AC3)

**Title: Observed Impacts of Aerosol Regimes on Energy and Carbon Fluxes in the Amazon Forest**

Response (blue color) to anonymous Referee #2 (black). The original manuscript was changed accordingly. The lines indicated in our answers correspond to the track version of the manuscript.

**General comments**

The work uses observations of AOD to evaluate impacts of aerosols on amazon forest energy balance fluxes at unique data set from a relatively new flux site in Manaus.

We would like to thank Referee #2 for their attention to detail and helpful comments, which have contributed to the improvement of the manuscript.

First of all we would like to begin our responses by stating that in the new version of the manuscript, we regrouped our data in a way that allowed us to include a greater number of runs (half-hour periods). In the previous version of the manuscript, in addition to excluding all periods when clouds were present, which is very common in the Amazon, we also excluded all data from a given day and time when a variable was missing. For example, if we did not have the reflected shortwave radiation measurement for a given time, we removed all other variables for that same time. This resulted in only 523 valid half-hour periods (370 dry season, 153 wet). In the new version of the Manuscript, we decided to regroup the variables so that they did not depend on each other. We first identified the periods in which we had the Clean and Polluted regimes and then identified how many runs of each variable were available for each regime. After this procedure, the number of runs increased substantially, as shown in Table R1, comparison between the dataset used in the first version of the manuscript (single database) and the dataset used for this new version (database by variable).

Table R1: Number of runs (half-hour periods) after all quality controls mentioned in section 2.2.

| Variables | Single database | | | | | Database by variable | | | | |
|---|---|---|---|---|---|---|---|---|---|---|
| | 10:00 -14:00 LT | | 07:00 -17:00 LT | | Total | 10:00 -14:00 LT | | 07:00 -17:00 LT | | Total |
| | No. Clean | No. Polluted | No. Clean | No. Polluted | No. Sample | No. Clean | No. Polluted | No. Clean | No. Polluted | No. Sample |
| $SWin(Wm^{-2})$ | 98 | 81 | 219 | 151 | 370 | 301 | 204 | 736 | 459 | 1195 |
| $SWout(Wm^{-2})$ | 98 | 81 | 219 | 151 | 370 | 301 | 204 | 736 | 459 | 1195 |
| $LWatm(Wm^{-2})$ | 98 | 81 | 219 | 151 | 370 | 301 | 200 | 733 | 453 | 1186 |
| $LWterr(Wm^{-2})$ | 98 | 81 | 219 | 151 | 370 | 301 | 204 | 735 | 459 | 1194 |
| $Rn(Wm^{-2})$ | 98 | 81 | 219 | 151 | 370 | 301 | 200 | 733 | 453 | 1186 |
| $H(Wm^{-2})$ | 98 | 81 | 219 | 151 | 370 | 197 | 192 | 455 | 389 | 844 |
| $LE(Wm^{-2})$ | 98 | 81 | 219 | 151 | 370 | 183 | 180 | 447 | 386 | 833 |
| $FCO_2(\mu molm^{-2}s^{-1})$ | 98 | 81 | 219 | 151 | 370 | 247 | 195 | 596 | 405 | 1001 |
| $G(Wm^{-2})$ | 98 | 81 | 219 | 151 | 370 | 301 | 218 | 741 | 487 | 1228 |

**Specific comments**

The work is highly relevant, and the data used is state of the art. However, most of the
analysis is done with output from a model rather than with the 30 min H and LE
observed fluxes (Figs 4-6). Justification for this approach was not 100% clear and there
is no mention of how good the models are at representing the observations and what
is the uncertainty related to the results inferred from such simulations. Why is not better
to use the data?
We thank the reviewer #2 for their comment. We clarify that all analyses in this study
were based on the measured data. The polynomial fit shown in Figs. 4-6 (Fig. 5-7 in
the new version of the manuscript) was applied solely as a smoothing technique for
visualization purposes. In the revised manuscript, we have included the 30-min
observed data points in the figures to better illustrate data variability. This clarification
has been incorporated into the manuscript as follows:
L139-141: "*To improve the visualization of the mean diurnal patterns, a 4th-order
polynomial curve was applied exclusively as a smoothing technique to the
observational data. This curve fitting was used solely for graphical purposes and does
not represent a physical or predictive model. All analyses were based on the measured
data.*"

[Figure]

Figure R1. Relationship between temperature and vapor pressure deficit (VPD) above the forest canopy
at the ATTO site for Clean and Polluted regimes during the dry season (2016–2022) (Figure 5 in the
new version of the manuscript).

[Figure]

Figure R2. Diurnal cycle of surface fluxes during the dry season (2016–2022) under Clean (blue) and
Polluted (red) regimes, highlighting the 10:00–14:00 LT period. Rn (net radiation), G (ground heat flux),
H (sensible heat flux), and LE (latent heat flux). (Figure 6 in the new version of the manuscript).

[Figure]

Figure R3. Diurnal cycle of $CO_2$ flux ($FCO_2$) during the dry season (2016–2022) under Clean (blue) and
Polluted (red) regimes, highlighting the 10:00-14:00 LT period (Figure 7 in the new version of the
manuscript).

The study estimates a cooling effect of 0.53C from aerosol on the forest-atmosphere interface. The authors should estimate what this means for forest surface temperature using the LW out fluxes, this is more relevant as the energy fluxes are driven by surface temperature rather than air temperature.

We thank the reviewer for this comment. In this study, forest surface temperature can be evaluated independently of LWout, because infrared surface temperature (Ts) was directly measured throughout the study period. Based on these measurements, mean Ts values were 32.6 ± 3.8 °C for the Clean regime and 31.7 ± 3.9 °C for the Polluted regime, indicating a surface cooling of 0.9 °C associated with aerosol conditions. For comparison, the corresponding air temperature difference between the two regimes was 0.3 °C.

The following sentence has been added to the revised manuscript.

L249-250: "*The cooling between the 10:00 and 14:00 LT regimes implies an average reduction in canopy surface temperature of 0.9 °C (not shown here), based on infrared surface temperature measurements, and a corresponding reduction in air temperature of 0.3 °C, resulting in a −2 hPa (13%) decrease in VPD.*"

In addition, Table 1 (in the revised manuscript) has been updated to include detailed information on the infrared surface temperature measurements.

Some parts of the work appear rather descriptive. Here two examples i) 247..the authors need to elaborate more specifically why/how this would lead to an increase in evapotranspiration

We thank the reviewer for this comment. We have revised the text to clarify the physiological mechanism linking enhanced $CO_2$ uptake and evapotranspiration. In addition, we estimated water-use efficiency (WUE) using $FCO_2/LE$ as a proxy to address the link between photosynthetic gas exchange and evapotranspiration, and revised the discussion accordingly.

L269-300: "*In the Polluted regime, $CO_2$ fluxes were more negative (Figure 7), indicating increased $CO_2$ uptake by vegetation related to photosynthetic activity. Such enhanced photosynthesis may be linked to changes in stomatal regulation that allow greater $CO_2$ uptake without a proportional increase in transpiration, reflecting higher stomatal conductance efficiency (Liu et al., 2022; Crous et al., 2025). However, analysis of the LE, which represents the fraction of available energy converted into evapotranspiration, shows a consistent decrease in the polluted regime compared to the Clean regime (Figure 6).*"

L302-307: "*The apparent paradox of an increase in $CO_2$ absorption alongside a*
*reduction in LE can be explained by differences in water use efficiency (WUE).*
*According to Dekker et al. (2016) and Yang et al. (2016), WUE is defined as the ratio*
*of carbon assimilated to water transpired by vegetation. In this study, WUE was*
*estimated using $FCO_2/LE$ as a proxy. WUE was significantly higher under Polluted*
*compared to Clean regime(mean values of 0.042 and 0.029, respectively, $p < 0.05$).*
*This indicates that under Polluted regime, vegetation assimilates more carbon per unit*
*of water lost, consistent with the observed reduction in latent heat flux (Figure 6)*
*despite enhanced $CO_2$ uptake (Figure 7).*

ii) Regarding impacts of aerosols on evapotranspiration and the relation to the CO2
enhancement, there is a key discussion missing around what happens to stomatal
conductance.

We agree with the reviewer and have revised the manuscript to address the role of
stomatal conductance in the discussion, as detailed in the response to the previous
comment (L106-123, in this document).

Line 35 The references in this line should come in parenthesis. Same in line 44

Thank you for pointing this out. The reference formatting in lines 35 and 45 has been
corrected in the revised manuscript.

236 -237 this sentence is unclear: 'The sum of H and LE was also found to be
67.85Wm−2 lower for the clean regime than for Rn,'

Thank you for pointing this out. We agree that the sentence was unclear. To address
this issue, we revised the text to avoid redundancy. A discussion of the surface energy
balance closure has been included (L278), and the sentence referring to the sum of H
and LE relative to Rn has been removed (L286) as follows:

L278-281: "*The surface energy balance closure was 0.89 for the clean regime and*
*0.88 for the polluted regime, comparable to values reported in the literature (Mauder*
*et al., 2024). The corresponding residuals were of similar magnitude (70 $Wm^{-2}$ for clean*
*and 75 $Wm^{-2}$ for polluted), indicating that the observed differences in energy fluxes are*
*not related to differences in energy balance closure.*"

L283-285: "*Sensible heat decreased by an average of -21.7 $Wm^{-2}$ (13.5 %), reflecting*
*reduced energy transfer to the atmospheric boundary layer. Similarly, LE decreased*
*by -8.9 $Wm^{-2}$ (2 %), indicating limited evapotranspiration due to the reduced radiative*
*energy available. The Bowen ratio, which relates H and LE, recorded 0.38 in the clean*
*regime and 0.33 in the polluted regime, suggesting that a higher proportion of energy*
*was allocated to latent processes, as expected in forest environments.*"

238-239: this could also be clearer : *It appears that the polluted regime is further from*
*the energy balance close, suggesting a change in how this energy is distributed.'*

Thank you for this comment. The sentence has been removed, as the revised
manuscript now includes a discussion of energy balance closure as detailed in
response to the previous comment.

Line 250 add units to VPD

The text has been revised accordingly. Thanks.

Line 255 water 'emitted' by evapotranspiration?

We thank the reviewer for pointing this out. We agree that the original wording was
imprecise. The text has been removed.

**References**

Crous, K. Y., Middleby, K. B., Cheesman, A. W., Bouet, A. Y., Schiffer, M., Liddell, M.
J., Barton, C. V., and Cernusak, L. A.: Leaf warming in the canopy of mature tropical
trees reduced photosynthesis due to downregulation of photosynthetic capacity and
reduced stomatal conductance, New Phytologist, 245, 1421–1436,
https://doi.org/10.1111/nph.20320, 2025.

Dekker, S. C., Groenendijk, M., Booth, B. B. B., Huntingford, C., and Cox, P. M.: Spatial
and temporal variations in plant water-use efficiency inferred from tree-ring, eddy
covariance and atmospheric observations, Earth System Dynamics, 7, 525–533,
https://doi.org/10.5194/esd-7-525-2016, 2016.

Liu, Y., Flournoy, O., Zhang, Q., Novick, K. A., Koster, R. D., and Konings, A. G.:
Canopy Height and Climate Dryness Parsimoniously Explain Spatial Variation of
Unstressed Stomatal Conductance, Geophysical Research Letters, 49,
https://doi.org/10.1029/2022GL099339, 2022.

Mauder, M., Jung, M., Stoy, P., Nelson, J., and Wanner, L.: Energy balance closure at
FLUXNET sites revisited, https://doi.org/10.1016/j.agrformet.2024.110235, 2024.

Yang, Y., Guan, H., Batelaan, O., McVicar, T. R., Long, D., Piao, S., Liang, W., Liu, B.,
Jin, Z., and Simmons, C. T.: Contrasting responses of water use efficiency to drought
across global terrestrial ecosystems, Scientific Reports, 6,
https://doi.org/10.1038/srep23284, 2016

---

## Author Comment (AC4)

**Title: Observed Impacts of Aerosol Regimes on Energy and Carbon Fluxes in the Amazon Forest**

Response (blue color) to anonymous Referee #3 (black). The original manuscript was changed accordingly. The lines indicated in our answers correspond to the track version of the manuscript.

**General comment**

This comment was prepared as part of MSc course work at Wageningen University under supervision of Prof Wouter Peters. They were uploaded as a comment as they were regarded to be of good quality, and likely helpful to the authors and editor in the review process.

We would like to express our sincere gratitude to Professor Wouter Peters and his students for their interest in our manuscript. Their comments help us to improve our results and discussion.

This study examines how aerosol regimes affect energy and carbon fluxes in a pristine central Amazon forest. Using 2016–2022 meteorological and flux data from the Amazon Tall Tower Observatory (ATTO) and AOD (500 nm) from AERONET, it tests whether aerosol loading alters latent heat (LE), net radiation (Rn), and $CO_2$ fluxes ($FCO_2$). The study focuses on the dry season (August–November), when biomass burning elevates aerosol concentrations across the southern Amazon Basin. The authors define two aerosol regimes: clean (AOD < 0.13) and polluted (AOD > 0.40), consistent with previous studies such as Steiner et al. (2013) and Ross Herbert & Stier (2023). This threshold-based approach, derived from data percentiles, provides a simple yet robust framework for distinguishing contrasting aerosol loading conditions. Their analysis focuses on the 10:00–14:00 LT period to examine energy partitioning under contrasting aerosol regimes. Authors interestingly present, VPD vs Temperature (Figure 4), a combination of variables that I have not encountered in other studies reviewed during this process. It is particularly valuable, as it effectively illustrates—through the observed variables of VPD and temperature—the realistic delay caused by reduced shortwave incoming radiation (SWin) during polluted periods. They report a delay in the rise of temperature and VPD under polluted conditions, highlighting the moderating effect of aerosols. They conclude by confirming the well-documented finding that, paradoxically, $CO_2$ uptake increases under polluted conditions—by about 57.7% in this case—due to the diffuse radiation effect, where scattered sunlight enhances photosynthesis within shaded canopy layers. This result is in strong agreement with previous studies on Amazonian aerosol dynamics, particularly Rodrigues et al. (2024) and Cirino et al. (2014), which similarly observed elevated carbon uptake under high-AOD conditions. The study concludes by emphasizing the
nonlinear and complex interactions between AOD and surface fluxes, demonstrated
through MANCOVA and Random Forest Model analyses, underlining however the
need for further investigation.
**Remarks on several aspects:**
(1) Midday Averaging
The authors assess the effects of aerosols on surface energy and carbon fluxes by
averaging 30-minute flux measurements over the 10:00–14:00 LT period and then
calculating percentage reductions between clean and polluted aerosol regimes. This
time window is identified as representing the period of strongest radiative and
convective activity (line 173). However, the diurnal cycle plots (Figs. 5 and 6) reveal
uneven flux patterns, with noticeable uninvestigated areas ( Figure 5 & 6 "white
spaces", outside 10:00–14:00 LT window) within both the clean and polluted regimes.
As a starting point, Figure 4 clearly shows a delay in the increase of temperature and
vapor pressure deficit (VPD). Because natural processes evolve non-uniformly
throughout the day, using a short and non-equidistant time subset which may bias the
calculated percentage reductions and misrepresent the actual aerosol influence.  The
paper's methodology follows Steiner et al. (2013), who also analyzed fluxes over the
10:00–14:00 LT period and compared similar aerosol optical depth (AOD) regimes
(AOD < 0.3 vs. > 0.5). However, within the text, fluxes reductions' comparisons are
made with studies that employed different approaches to assess aerosol-load effects.
For example, Rodrigues et al. (2024) and Cirino et al. (2014) estimated flux reductions
under specific irradiance conditions, distinguishing Solar Zenith Angle (SZA) zones
and thereby incorporating the time-of-day variability, rather than relying on a fixed
midday average. A closer examination of Figures 5 and 6, which depict sensible, latent,
and ground heat fluxes, reveals an interesting but unexamined pattern during i.e. the
morning transition (06:00–10:00 LT). Both H and G occasionally exceed their
respective values under polluted conditions, while the consistent dominance of the
clean regime in LE appears to be underestimated. Early-morning $CO_2$ uptake (Figure
6) also exhibits a more dynamic behavior, with pronounced transitions between clean
and polluted regimes. To better capture the full evolution of the phenomena and
associated fluxes, the authors could integrate the area under the fluxes' curves over
the 06:00–17:00 LT period and compare the resulting averages between the clean and
polluted aerosol regimes. Alternatively, if there is sufficient data outside the window
10:00-14:00 LT the authors could consider reporting morning (06:00-10:00 LT) and
afternoon sub-period (14:00-17:00 LT) averages separately to capture diurnal
variability better. Analyzing relative irradiance would require substantially more
methodological development and investigation by the authors; therefore, it is not
recommended.

We thank the Referee #3 for this comment. We agree that surface fluxes exhibit
variable features outside the 10:00–14:00 LT window. However, during these periods,
flux variability may be influenced by boundary-layer dynamics and low solar elevation
angles, which can affect H, LE, and $FCO_2$ and complicate the isolation of the radiative
effects of aerosols. Moreover, radiometer uncertainties (typically within ~5%) are less
significant when radiation levels are high. At low solar elevation angles (early morning
and late afternoon), radiation magnitudes are smaller, which increases the relative
importance of measurement errors and energy balance closure uncertainties. For
these reasons, the 10:00–14:00 LT period provides more favorable conditions for
isolating aerosol-induced radiative effects.

In the revised manuscript, we have added Figure 4 showing the full diurnal cycles of
shortwave, longwave, and net radiation during the dry season (2016–2022). This figure
demonstrates that peak net radiation consistently occurs between 10:00 and 14:00 LT,
supporting our choice of this time window.

(2) Gaps Manipulation

The authors state that their initial dataset comprised 10,890 half-hourly observations
(line 87), which, after several filtering steps, was reduced to 523 rows—of which only
370 belong to the dry season (lines 94–96). However, the paper does not clarify how
these 10,890 records were originally obtained. Figure 2 further raises questions about
data representativeness and statistical treatment: the monthly boxplots show means
much higher than medians, indicating positive skewness, while the number of valid
data points per month is not reported. The data filtering process is clearly described,
resulting in 523 rows of 30-minute averaged meteorological, flux, and AOD values.
However, the dataset distribution across years is highly uneven, as also noted by the
authors (line 97: "The distribution…effects of aerosol"). Specifically, years contributing
less than 5 % of the total dataset are treated equivalently to years such as 2020 and
2022 (42,4% and 29,2% data coverage respectively), despite potentially different
atmospheric and surface conditions. This raises concerns regarding the robustness of
the study's conclusions. Evidentially, no quantitative assessment of data
representativeness or uncertainty is provided. Similar studies (e.g., Schmitt et al.,
2023) have explicitly visualized monthly data availability and included "fraction of
missing data". Moreover, the extremely low number of data rows for certain years
warrants further examination, as such sparse temporal coverage could substantially
affect the robustness of the Random Forest Model (RFM) used later in the statistical
analysis. Limited data availability may lead to overfitting, biased feature importance
when training and validation subsets are unevenly represented. It is recommended that
the authors include the fraction of valid rows per month, which could be directly
incorporated into Figure 2. Furthermore, the manuscript should clearly describe the origin of the initial 10,890 observations—specifying the time period covered, sampling frequency, and measured variables—to better contextualize the subsequent data filtering process.

We thank the Referee #3 for these important comments. They will certainly help to improve the methodology and discussion of the results.

We would like to begin our responses by stating that in the new version of the manuscript, we regrouped our data in a way that allowed us to include a greater number of runs (half-hour periods). In the previous version of the manuscript, in addition to excluding all periods when clouds were present, which is very common in the Amazon, we also excluded all data from a given day and time when a variable was missing. For example, if we did not have the reflected shortwave radiation measurement for a given time, we removed all other variables for that same time. This resulted in only 523 valid half-hour periods (370 dry season, 153 wet). In the new version of the Manuscript, we decided to regroup the variables so that they did not depend on each other. We first identified the periods in which we had the Clean and Polluted regimes and then identified how many runs of each variable were available for each regime. After this procedure, the number of runs increased substantially, as shown in Table R1, comparison between the dataset used in the first version of the manuscript (single database) and the dataset used for this new version (database by variable).

Table R1: Number of runs (half-hour periods) after all quality controls mentioned in section 2.2.

| Variables | Single database | | | | | Database by variable | | | | |
|---|---|---|---|---|---|---|---|---|---|---|
| | 10:00 -14:00 LT | | 07:00 -17:00 LT | | Total | 10:00 -14:00 LT | | 07:00 -17:00 LT | | Total |
| | No. Clean | No. Polluted | No. Clean | No. Polluted | No. Sample | No. Clean | No. Polluted | No. Clean | No. Polluted | No. Sample |
| $SWin(Wm^{-2})$ | 98 | 81 | 219 | 151 | 370 | 301 | 204 | 736 | 459 | 1195 |
| $SWout(Wm^{-2})$ | 98 | 81 | 219 | 151 | 370 | 301 | 204 | 736 | 459 | 1195 |
| $LWatm(Wm^{-2})$ | 98 | 81 | 219 | 151 | 370 | 301 | 200 | 733 | 453 | 1186 |
| $LWterr(Wm^{-2})$ | 98 | 81 | 219 | 151 | 370 | 301 | 204 | 735 | 459 | 1194 |
| $Rn(Wm^{-2})$ | 98 | 81 | 219 | 151 | 370 | 301 | 200 | 733 | 453 | 1186 |
| $H(Wm^{-2})$ | 98 | 81 | 219 | 151 | 370 | 197 | 192 | 455 | 389 | 844 |
| $LE(Wm^{-2})$ | 98 | 81 | 219 | 151 | 370 | 183 | 180 | 447 | 386 | 833 |
| $FCO_2(\mu molm^{-2}s^{-1})$ | 98 | 81 | 219 | 151 | 370 | 247 | 195 | 596 | 405 | 1001 |
| $G(Wm^{-2})$ | 98 | 81 | 219 | 151 | 370 | 301 | 218 | 741 | 487 | 1228 |

The initial number of 10,890 observations does not represent the full raw eddy-covariance dataset, which contains 122,734 half-hourly records over 2016–2022. Instead, this number corresponds to the subset of 30-minute periods for which aerosol optical depth (AOD) data from AERONET (version 3, level 2) were available and could be matched with surface flux and radiation measurements. The text has been updated accordingly to improve clarity (Section 2.2 in the revised version of the manuscript).

L95-107: "*To eliminate cloud interference and investigate the role of aerosols in surface energy fluxes, the central objective of this study, we used data from the Aerosol Robotic Network (AERONET) at the ATTO site, specifically AOD (version 3, level 2). These data are free of cloud contamination due to pre and post-field calibration (Giles et al., 2019). Based on this, 30-minute averages were calculated between 2016 and*

*2022 for which AOD data from AERONET were available, the initial combined dataset comprised 10,890 observations, including all variables listed in Table 1. This matched dataset served as the starting point for the subsequent quality control and filtering procedures. First, the turbulent fluxes underwent quality control based on Foken et al. (2004). Only data with flags "0" (best quality) and "1" (acceptable for general analysis) were used; data with flag "2" (poor quality) were discarded. Second, this study only considered the daytime period (from 7:00 to 17:00 LT) because the highest Rn values occur during this time. After filtering, the resulting dataset is summarized in Table S1 and S2."*

As described in the previous comment, we regrouped our data in a way that allowed us to include a greater number of runs (half-hour periods). Based on this updated dataset, Figure R1 was revised and now includes the number of samples per month ($n$). The mean values are higher than the medians, particularly during the dry season, reflecting the influence of episodic high-AOD events (e.g., biomass burning, smoke intrusions) that shift the distributions toward positive skewness. We additionally verified that the main seasonal contrasts remain qualitatively unchanged when using median AOD instead of mean AOD.

[Figure]

Figure R1. Box plot showing monthly AOD 500 nm values measured at the ATTO site between 2016 and 2022. The box represents the central 50% of the data, the whiskers represent the smallest and largest non-outlier values, while the means are indicated by the green triangles and the medians are the lines inside the box. Numbers above each month indicate the sample size (n) (Figure 2 in the revised version of the manuscript).

(3) Statistical Analysis

The study explores the relationship between aerosol optical depth (AOD) and surface fluxes (Rn, H, LE, $FCO_2$) implementing Spearman correlations, multivariate MANCOVA testing assessed by Pillai test and a Random Forest Model (RFM) to quantify nonlinear dependencies and variable importance. However, several methodological lack in processes or data-handling limitations seem to weaken the robustness of the conclusions. The manuscript provides a general introduction to the application of Pillai's test and outlines the advantages of using the Random Forest Model (RFM) to investigate nonlinear and complex interactions between variables and systems. However, it remains unclear to what extent these principles—particularly in the case of RFM—have been appropriately implemented and demonstrated in the study. In comparable RFM environmental works, such as Miao et al. (2018) and Zhang et al. (2023), linear correlation analyses were explicitly conducted to assess collinearity among key variables by providing comprehensive correlation matrices, providing direct linear insights. In contrast, Rocha et al. (2025) only briefly mention in line 272 that "the statistical relationships show low intensity or no statistical significance," without offering supporting analyses or graphical evidence. Furthermore, while Miao et al. (2018) thoroughly examined their multivariate equations and reported the statistical significance of their models and variables, Rocha et al. (2025) limit the discussion to the significance of Pillai's test (line 275), suggesting the absence of linear interactions without presenting sufficient analytical support or methodological transparency. Another major concern is data volume, as mentioned in major argument 2. Miao et al. (2018) utilized approximately 7,000 samples, and Zhang et al. (2023) worked with about 60,000 samples. In contrast, Rocha et al. (2025) rely on only 370 rows of data for the dry period, which raises serious concerns about potential overfitting of the RFM. Moreover, although the manuscript mentions a cross-validation approach in Table 3, it does not specify the technique used or report its results. Finally, the model assessment presented in Table 3 appears inadequate and leaves substantial uncertainty regarding the RFM's reliability. In the referenced studies, Miao et al. (2018) implemented multiple factor matrices, and Zhang et al. (2023) validated their models through scatter density plots and strong statistical metrics across training and testing datasets, including mean absolute error (MAE) and percentage variation analyses. Rocha et al. 2025 attempt to employ a RFM to capture the nonlinear influence of aerosols on surface fluxes. However, this approach lacks sufficient methodological justification and statistical robustness. The authors do not provide any evidence of cross-validation or other procedures to assess model generalisability. Furthermore, the dataset used for training—only 370 observations—is several orders of magnitude smaller than what is typically required for stable Random Forest performance, raising serious concerns about overfitting and the reliability of the reported metrics. Consequently, the predictive results presented in Table 3 should be interpreted with caution, as their statistical validity is uncertain. Given the limited dataset, the application of the Random Forest Model (RFM) in this study does not appear to add substantial value to the results or discussion. With such a small sample size, the model's capacity to generalise is minimal, and its predictive performance cannot be reliably validated. Moreover, the manuscript provides no detailed explanation of the model evaluation or validation procedures, which further undermines confidence in the reported outcomes. To strengthen the analysis, I suggest replacing or complementing the RFM with a correlation matrix to explicitly reveal potential collinearity among variables, particularly regarding the influence of AOD (as in Table 3). Additionally, presenting multivariate regression equations and reporting their levels of statistical significance would offer a clearer and more interpretable understanding of how other environmental factors interacts with AOD. Such an approach could also serve as a solid foundation for future studies investigating aerosol impacts on surface fluxes under polluted regimes.

We sincerely thank the MSc students at Wageningen University, under the supervision of Prof Wouter Peters, for their detailed and constructive feedback regarding our statistical methodology.

We agree with the referee and have removed the RFM analysis from the revised manuscript. We emphasize that the RFM was originally intended as a complementary exploratory tool, and its removal does not affect the main results or interpretations of the study. In the revised version, to assess whether clean and polluted regimes exhibit statistically significant differences in radiation and surface energy and $CO_2$ fluxes, we apply the Mann–Whitney U test, which is well suited for non-normally distributed data and unequal sample sizes. These revisions provide a clearer and more robust statistical framework to support our conclusions regarding aerosol impacts on surface–atmosphere interactions.

**Minor arguments and typos:**

**Minor issue 1**: Several sentences are poorly structured or ambiguous, leading to confusion or misinterpretation. Examples include lines 74–75, 97, 99–101, 112–113, 134, and 247–248, as well as the descriptions for Figures, especially 2 and 4, where I suggest rephrasing or clarifying.

L76: "*The climate is tropical humid and characterized by two seasons (wet and dry), driven by seasonal shifts of the Intertropical Convergence Zone over the Amazon Basin (Andreae et al., 2015).*"

L299: "*However, analysis of LE, which represents the fraction of available energy converted into evapotranspiration, shows a consistent decrease in the polluted regime compared to the clean regime (Figure 6), which contradicts this expectation.*"

Additionally, Section 2.3 (Analysis Methods) has been revised to address all the reviewer's comments.

**Minor issue 2**: Several statements lack adequate justification or references, I suggest further elaboration on the statements:

Line 114: The use of a fourth-order polynomial is mentioned but not explained or visualized.

The polynomial fit shown in Figs. 4–6 was applied solely as a smoothing technique for visualization purposes. In the revised manuscript, we have included the 30-min observed data points in the figures to better illustrate data variability. This clarification has been incorporated into the manuscript as follows:

L139-141: "*To facilitate the visualization of the mean diurnal patterns, a 4th-order polynomial curve was applied exclusively as a smoothing technique to the observational data. This curve fitting was used solely for graphical purposes and does not represent a physical or predictive model. All analyses were based on the measured data.*"

Lines 135–136: Require citation or elaboration.

We appreciate your suggestion, but we have removed the RFM.

Lines 220–222: Could be expanded with a brief example of the described method.

The expansion was carried out as follows:

L261-267: "*They identified a correlation between relative irradiance, air temperature, and VPD. Meanwhile, Herbert and Stier (2023) and Palácios et al. (2024) reinforce the idea that AOD significantly influences temperature variations, particularly on a regional scale. For instance, Palácios et al. (2024) observed positive linear correlations between AOD and air temperature across distinct climatic phases, attributed to the absorption of solar radiation by biomass burning emissions resulting in atmospheric heating. Similarly, Herbert and Stier (2023) utilized reanalysis data to demonstrate that 2-meter air temperature increases as a function of AOD, consistent with localized heating of the smoke layer due to strong absorption of solar radiation.*"

**Minor Issue 3**: Some methodological descriptions (e.g., line 12 in the Abstract; lines 87–90 on data filtering; lines 137–144 on the RFM methodology) could be condensed, as they do not add substantial value to the manuscript.

We thank the referee for this comment. The RFM analysis has been removed from the revised manuscript, as detailed in lines 251-259 of this document.

**Minor Issue 4**: GPP is mentioned in the Abstract and Conclusion but is neither discussed nor analyzed in the main text.

The Abstract and Conclusion has been updated, and references to GPP have been removed in the revised manuscript.

**Minor Issue 5**: The manuscript refers to two towers at the ATTO site but does not specify which tower's data are used in the analyses and figures.

Thanks for this comment. The text has been revised to specify that the analyses are based on data from the Instant Tower (81 m).

L66: "*The data used in this study were collected as part of the ATTO project, a bilateral initiative between Brazil and Germany. Since 2012, ATTO has carried out continuous measurements, as described by Andreae et al. (2015), located in an area of pristine tropical forests in the central Amazon (Figure 1), which contains the Instant Tower of 81 meters (-2.1441°S, -58.9999°W).*"

P1, line 12: The last sentence of the Abstract adds no clear value to the manuscript and could be removed.

The text has been removed from the abstract in the new version of the manuscript.

P3, line 81: Change LiCor to LI-COR for correct company citation.

The text has been updated accordingly. Thanks.

P5, line 112: The text states that hourly averages are used, while figures show 30-minute values—this inconsistency should be corrected.

Section 2.3 (Analysis Methods) has been revised accordingly.

P14, Table 3 description: Typo — change FCO to $FCO_2$.

The table description has been updated accordingly. Thanks.

P15, line 312: Typo — change aerossol to aerosol.

The text has been updated accordingly. Thanks.

References:

[revised manuscript text omitted]